# Evaluating Outer Membrane Vesicle Isolation Techniques for *Borrelia burgdorferi* and Their Impact on Vesicle Composition, Gene Expression Profile and Uptake

**DOI:** 10.3390/antibiotics14111079

**Published:** 2025-10-27

**Authors:** Jasmine Jathan, Jay M. Pandya, Mahima Jain, Tejasri Kaithalapuram, Dhara Cherukuri, Eva Sapi

**Affiliations:** Lyme Disease Research Group, Department of Biology and Environmental Science, University of New Haven, 300 Boston Post Road, West Haven, CT 06516, USA; jjath1@unh.newhaven.edu (J.J.); jpand2@unh.newhaven.edu (J.M.P.); mjain8@unh.newhaven.edu (M.J.); tkait1@unh.newhaven.edu (T.K.); dcher4@unh.newhaven.edu (D.C.)

**Keywords:** Lyme disease, *Borrelia burgdorferi*, bacterial virulence factors, outer membrane vesicles (OMVs), host-pathogen interactions, OMV isolation, mammalian cell uptake

## Abstract

**Background**: *Borrelia burgdorferi*, the causative agent of Lyme disease, releases outer membrane vesicles (OMVs) that may contribute to infection and modulate the host immune response. Although interest in OMVs is growing, few studies have systematically compared methods for isolating OMVs from *B. burgdorferi*. **Methods**: In this study, we evaluated two OMV isolation techniques—standard ultracentrifugation and an ion-exchange chromatography-based ExoBacteria™ kit—and examined how serum supplements (rabbit serum vs. exosome-depleted fetal bovine serum, ED-FBS) influence Bb-OMV yield and composition. Gene expression profiles were assessed using RT-PCR, and specific protein content was identified by Western blot analyses. To assess the ability of Bb-OMVs to interact with host cells, Bb-OMVs were co-cultured with MDA-MB-231 triple-negative breast cancer cells. **Results**: Transmission electron microscopy confirmed that both methods produced spherical Bb-OMVs with intact membrane bilayers. Ultracentrifugation generated larger vesicles (15–180 nm), while the ExoBacteria™ kit yielded smaller vesicles (<50 nm) with a higher double-stranded DNA (dsDNA) content, and protein levels were similar across samples. Cultures grown with rabbit serum produced more Bb-OMVs and had cleaner backgrounds in the TEM images than those grown with ED-FBS. All Bb-OMV samples lacked intracellular markers (DnaK and 16S rRNA) and consistently expressed the outer surface protein OspA, confirming high purity. All isolated Bb-OMVs were taken up by the cells, as indicated by OspA expression, without detectable 16S rRNA, confirming vesicle internalization without bacterial contamination. **Conclusions**: These findings indicate that isolated OMVs are biologically active and capable of interacting with mammalian cells, highlighting their potential role in host–pathogen interactions and the broader relevance of OMVs in studying bacterial modulation of mammalian cell behavior. Overall, both isolation methods produced high-quality OMVs, with ultracentrifugation yielding slightly more pure vesicles, emphasizing the importance of selecting appropriate isolation methods and culture conditions for functional OMV studies.

## 1. Introduction

Bacteria communicate between themselves and the environment in many ways, one of which is through the secretion of outer membrane vesicles [1,2,3,4]. Gram-negative bacteria can naturally secrete OMVs throughout their growth phase and occasionally in response to certain inductions like temperature, pH, and nutrients in their environment [5,6,7]. OMVs are lipid vesicles produced due to the blebbing of the bacterial phospholipid outer membrane or by explosive cell lysis [8,9,10]. These OMVs are spherical-like structures with sizes ranging from 11–200 nm and can vary in the contents and amounts of cargo molecules like DNA, RNA, proteins, and endotoxic lipopolysaccharides [11,12].

Previous research has demonstrated that *Borrelia burgdorferi*, the causative agent of Lyme disease, releases OMVs that potentially contain virulence factors such as outer surface proteins (Osps) A, B, and C, as well as DNA, enabling these vesicles to adhere to human endothelial cells [13,14,15,16,17]. It has been suggested that these virulence factors play a significant role in *B. burgdorferi* infection and in modulating host cell functions, which may further contribute to the pathogenesis of Lyme disease [17,18].

*B. burgdorferi* is an obligate extracellular spirochete with a complex life cycle alternating between Ixodes tick vectors and mammalian hosts [19]. In ticks, spirochetes reside in the midgut and are transmitted to mammals during blood feeding [19]. Once in the mammalian host, *B. burgdorferi* disseminates through the bloodstream and extracellular matrices, colonizing multiple tissues. During this process, spirochetes can invade various cell types, including endothelial, epithelial, and tumorigenic mammary epithelial cells, facilitating persistence, evasion of host immune responses, and chronic infection [20,21,22,23,24,25,26]. Notably, *B. burgdorferi* produces OMVs throughout its life cycle, which are thought to play a role in host–pathogen interactions by delivering bacterial proteins and nucleic acids to host cells, modulating immune responses, and potentially aiding in colonization and survival [13,14,15,16,17,18]. Understanding these dynamics provides critical context for investigating the functional relevance of OMVs in infection and host tissue interactions.

Despite growing interest in OMVs and their role in bacterial pathogenicity, relatively few studies have systematically compared OMV isolation methods in *B. burgdorferi* [15,27,28]. Most studies have relied on ultracentrifugation [13,14,15,16,17,27,29], which, while effective, requires specialized equipment that may not be accessible in smaller laboratories or universities. Systematic evaluation of different isolation techniques is critical because the choice of method can influence OMV yield, purity, and cargo composition, ultimately affecting downstream functional analyses [30]. To address this gap, in the present study, we evaluated two OMV isolation techniques: the standard ultracentrifugation method and a commercially available ExoBacteria™ OMV isolation kit (System Biosciences (SBI), Palo Alto, CA, USA). By comparing these approaches under controlled conditions, we aimed to identify reliable and accessible methods for producing biologically active OMVs suitable for functional studies in mammalian cell models. We also investigated the effect of OMV production by culturing *B. burgdorferi* with the standard rabbit serum and exosome-depleted fetal bovine serum (ED-FBS).

The isolated Bb-OMVs were characterized using transmission electron microscopy (TEM) to assess their size and morphology. In addition, we analyzed their DNA and protein content and evaluated the gene expression profiles of Bb-OMV samples using RT-qPCR and Western blotting techniques. Finally, we investigated the ability of the isolated Bb-OMVs to enter mammalian epithelial cells to further understand their potential role in host–pathogen interactions. This comparative analysis aims to identify the most effective and reliable method for isolating Bb-OMVs from *B. burgdorferi*, thereby facilitating future studies on their biological significance and potential as diagnostic or therapeutic targets.

## 2. Results

### 2.1. Identification of Bb-OMVs Derived from B. burgdorferi

Outer membrane vesicles (OMVs) released from log-phase *B. burgdorferi* B31 strain were purified using two approaches: ultracentrifugation and ion-exchange chromatography-based ExoBacteria™ kit. The isolated Bb-OMVs were negatively stained with 2% uranyl acetate and imaged using high-resolution transmission electron microscopy (TEM). TEM analysis revealed that Bb-OMVs obtained from both purification methods exhibited an intact, continuous single bilayer membrane with spherical morphology (Figure 1A–D). The sizes of purified *B. burgdorferi* OMVs from each isolation condition were analyzed from the transmission electron micrographs, by measuring the diameters of ~300 vesicles using ImageJ2 (version 2.16.0) (Figure 2A,B). Overall, the diameters of the purified Bb-OMVs were comparable between each condition, varying from 11–200 nm, with the majority of the Bb-OMVs being between 20–60 nm. Purification by ultracentrifugation was observed to produce more Bb-OMVs larger than 50 nm compared to purification by the ExoBacteria™ kit. To assess the influence of serum-derived extracellular vesicles on OMV production, cultures were also grown in BSK-H medium supplemented with 6% rabbit serum (standard medium for *B. burgdorferi* cultures). While all primary experiments used exosome-depleted FBS (ED-FBS), the inclusion of rabbit serum provided a comparative condition. Cultures supplemented with rabbit serum produced a slightly greater number of Bb-OMVs and exhibited cleaner backgrounds in TEM images compared to those supplemented with ED-FBS. Using rabbit serum as a stimulation serum yielded a higher number of smaller Bb-OMVs, predominantly under 50 nm in diameter. These observations are consistent with prior reports indicating that OMVs derived from Gram-negative bacteria form spherical bilayer membrane structures ranging in size from 11 to 250 nm [17,27]. Comparing the number of Bb-OMVs isolated between the ultracentrifugation method and ExoBacteria™ kit, significant differences were found in the less-than-50 nm (*p* value < 0.01) and greater-than-100 nm (*p* value < 0.001) size ranges when 10% ED-FBS was used as a stimulation serum.

### 2.2. Characterization of OMVs Derived from B. burgdorferi

The isolated Bb-OMVs were evaluated for the presence of double-stranded DNA (dsDNA) and total protein content to confirm cargo composition and compare isolation methods. For dsDNA quantification, bovine serum albumin (BSA, fraction V) was used as a negative control to exclude background DNA signals, while purified *B. burgdorferi* genomic DNA was included as a positive control standard. All purified Bb-OMV preparations contained detectable amounts of dsDNA, confirming that nucleic acids are packaged within the vesicles. Notably, the dsDNA concentrations and overall yield were found to be significantly higher in Bb-OMVs isolated using the ExoBacteria™ compared to ultracentrifugation (Figure 3A,B, *p*-value < 0.01).

Protein content was also quantified across the Bb-OMV samples. For protein quantification, BSA was used as a positive control to validate protein detection, while *B. burgdorferi* genomic DNA was used as a negative control to exclude non-protein signals. The protein concentrations measured in all Bb-OMV preparations were comparable to those observed in *B. burgdorferi* whole-cell lysate (Figure 4A), indicating that the vesicle preparation retains substantial protein content. However, total protein yield was significantly higher in Bb-OMVs purified using the ExoBacteria™ kit regardless of the serum type used for stimulation (rabbit serum or ED-FBS, Figure 4B, *p*-values < 0.01). For reference, total protein isolated from approximately 100 million *B. burgdorferi* spirochetes and 1 mg/mL albumin were used (Figure 4).

### 2.3. Analysis of Gene and Protein Expression Profiles of OMVs from B. burgdorferi Using RT-PCR and Western Blot Analyses Method

The gene expression profiles of Bb-OMV preparations were evaluated by assessing the presence of 16S rRNA (a bacterial chromosomal gene) expression, which is an indicator of *B. burgdorferi*-specific intracellular components, and a *B. burgdorferi*-specific outer surface protein, OspA, using RT-PCR. Genomic DNA from *B. burgdorferi* strain B31 was included as a positive control. Quantification and statistical analysis of results from three independent experiments demonstrated that Bb-OMVs purified by either ultracentrifugation or the ExoBacteria™ kit showed no detectable signal for 16S rRNA, indicating the absence of contaminating bacterial genomic material in the vesicle preparations. All Bb-OMV samples consistently expressed the OspA gene, regardless of the serum type used for stimulation (rabbit serum or ED-FBS, Figure 5). These results demonstrate that purified Bb-OMVs are free of detectable chromosomal DNA contamination and retain *B. burgdorferi*-specific outer surface protein marker.

To further assess the protein expression profiles of the different Bb-OMV samples, we used SDS-PAGE and Western blot analysis to evaluate the presence of the *B. burgdorferi*-specific outer surface protein A (OspA) and the absence of the intracellular heat shock protein DnaK. These markers were selected based on previous studies demonstrating their reliability in distinguishing Bb-OMVs from whole-cell contaminants [15,16,17]. Whole-cell lysates of *B. burgdorferi* were included as a positive control.

Western blot results showed that *B. burgdorferi* whole-cell lysates expressed both OspA and DnaK, as expected. In contrast, none of the Bb-OMV samples displayed detectable levels of the 70 kDa DnaK protein, confirming the absence of cellular contamination. All Bb-OMV preparations tested positive for OspA, with the strongest expression observed in Bb-OMVs isolated via ultracentrifugation, while weaker expression was noted in Bb-OMVs isolated using the ExoBacteria™ kit and from samples prepared with ED-FBS (Figure 6). Notably, signal levels were especially low in the ExoBacteria kit with ED-FBS, approaching the limit of detection and potentially reflecting background or non-specific binding.

### 2.4. Functional Analyses of B. burgdorferi OMVs Presence in OMV-Exposed Breast Cancer Cells Using RT-PCR

It was previously demonstrated that *B. burgdorferi*’s spirochetes can invade breast cancer cells as early as 24 h [25,26]. Here we evaluated whether Bb-OMVs isolated by different methods are also capable of entering epithelial cells. The triple-negative breast cancer cell line MDA-MB-231 was selected because previous studies, including our own, have shown that it efficiently internalizes *B. burgdorferi* and bacterial OMVs, making it an ideal model to study host–pathogen interactions and OMV uptake in mammalian epithelial cells [25,26,27].

The expression of 16S rRNA and the surface protein OspA in MDA-MB-231 cells infected with *B. burgdorferi* spirochetes or co-cultured with the different Bb-OMV samples were analyzed using RT-qPCR. The RNA was extracted from uninfected MDA-MB-231 cells and those cells co-cultured with *B. burgdorferi* spirochetes or Bb-OMVs for 24, 48, and 72 h. Results from three independent experiments demonstrated that *B. burgdorferi* efficiently invades breast cancer cells directly infected by spirochetes, as evidenced by the presence of both 16S rRNA and OspA expression at all time points (Figure 7A–C). In contrast, breast cancer cells exposed to *B. burgdorferi* OMVs showed no detectable 16S rRNA, confirming the absence of any spirochetal signature, but did exhibit positive OspA expression, indicating that Bb-OMVs are capable of entering the cells (Figure 7A–C). The results were compared between the OMV isolation methods using the same stimulation serum. There were no statistically significant differences (*p*-value > 0.05) observed in the expression of OspA in breast cancer cells exposed to different OMV sample preparations (Figure 7A–C). At the 48 h time point, MDA-MB-231 cells exposed to Bb-OMVs stimulated with rabbit serum showed a significant 4-fold difference in OspA expression when comparing ExoBacteria kit to the ultracentrifuge method (*p*-value < 0.05) (Table 1).

## 3. Discussion

The accurate isolation and characterization of outer membrane vesicles (OMVs) from *B. burgdorferi* are essential for advancing our understanding of their biological relevance, particularly in the context of infection and disease processes [15,16,17,18,28]. Although the ultracentrifuge method remains an effective method of OMV isolation [13,14,15,16,17,27,29], it also requires specialized equipment, which smaller institutions might not have access to. Therefore, it is important to test different scientific methods for OMV isolation as this choice can affect various factors like the OMV purity, yield, and cargo composition, ultimately affecting downstream processes [30]. Thus, in this study, we compared two OMV isolation methods, ultracentrifugation and an ion-change chromatography-based ExoBacteria™ kit, and evaluated how the choice of stimulation serum (rabbit serum vs. exosome-depleted fetal bovine serum) influences OMV yield, composition, and function. Table 2 provides a comprehensive overview of the different methods, along with the different experimental conditions used for OMV isolation in this study.

Using high-resolution TEM, we confirmed that Bb-OMVs isolated via both ultracentrifugation and the ExoBacteria™ kit displayed the expected spherical morphology with continuous single-membrane bilayers [8,9]. However, the size distribution of vesicles varied: ultracentrifugation produced a wider range of vesicles (11–180 nm, higher number of vesicles > 50 nm), while the ExoBacteria™ kit yielded a greater proportion of smaller Bb-OMVs (<50 nm). These results align with previous reports by Karvonen et al., who found naturally formed *B. burgdorferi* OMVs to range between 11 and ~108 nm, with an average diameter of 33 nm [17]. They also noted that previously published chemically induced OMV preparations were abnormally large (up to 1000 nm), further supporting the importance of natural isolation methods [17].

Furthermore, our findings are also in agreement with a recent technical study in which ultracentrifugation with 6% rabbit serum was used to obtain Bb-OMVs with a size range of 30–100 nm. The study reported robust vesicle yields and consistent morphology [29].

Another important factor to consider for OMV release from bacterial cells is determining the optimal harvest time based on the growth curve and the viability of the culture. Cultivating bacteria into the extremely late stationary phase can increase OMV yields, but it may also cause cell lysis, resulting in contamination from disrupted membranes and cytoplasmic proteins [31]. In the case of *Francisella novicida*, it was observed that the OMV yield was higher during early stationary phase than the mid-logarithmic phase, with differences in their protein profiles [32].

Numerous studies have shown that these vesicles contain a wide range of molecular cargo, including outer surface proteins, DNA, and RNA transcripts, starting with an early observation by Garon et al., who identified tightly packed DNA within *B. burgdorferi* OMVs [15,17,27,33].

Our study measured significant amounts of dsDNA and protein contents in all Bb-OMV samples. Interestingly, Bb-OMVs isolated using the ExoBacteria™ kit exhibited higher dsDNA content, while total protein levels remained comparable across all preparations. This differential cargo distribution reinforces the idea that different isolation methods may enrich distinct Bb-OMV subpopulations, an important consideration when studying OMV-mediated signaling or host interactions [27,28].

The results obtained showed that the choice of serum supplement also influenced Bb-OMV production. Cultures grown with rabbit serum produced slightly higher Bb-OMV yields and cleaner TEM backgrounds compared to those grown in ED-FBS. This supports the notion that rabbit serum may promote Bb-OMV formation, but also underscores the risk of contamination, which must be carefully addressed when interpreting Bb-OMV content and function [14,27].

To confirm Bb-OMV identity and gene expression profile, we assessed several markers such as OspA as a positive marker, 16S rRNA as a negative control, and DnaK as an additional control using Western blot and RT-PCR analyses. All Bb-OMV preparations lacked the intracellular marker DnaK and the bacterial chromosomal 16S rRNA signal, indicating high purity. In contrast, OspA, a well-established marker and one of the most consistently identified proteins for *B. burgdorferi* in previous studies [15,16,17], was detected in all samples, with stronger expression in ultracentrifugation-derived Bb-OMVs. These results further validate our use of OspA as a marker for Bb-OMV tracking and support its functional relevance in downstream studies.

To explore the biological relevance of Bb-OMVs, we assessed their uptake by MDA-MB-231 triple-negative breast cancer cells. Different serum supplements, such as rabbit serum versus exosome-depleted fetal bovine serum (ED-FBS), can affect Bb-OMV yield, size, and cargo composition, including DNA and protein content. These differences may, in turn, modulate OMV uptake by mammalian cells and their functional effects on host–pathogen interactions. By highlighting these influences, we emphasize the importance of carefully selecting culture conditions to ensure consistent and biologically relevant OMV preparations for functional studies [17,27,30].

Proper OMV isolation is critical for accurately assessing their biological effects, particularly in cancer research [34,35]. Contaminants such as free bacterial proteins, nucleic acids, or cellular debris can confound functional assays, leading to misleading conclusions about OMV-mediated effects on cancer cell behavior. By carefully selecting and validating isolation methods, researchers can ensure that observed interactions, such as uptake, signaling, or modulation of host cell pathways, are specifically attributable to OMVs. This systematic approach is essential for understanding their potential role in tumor progression, host–pathogen interactions, and the development of OMV-based diagnostic or therapeutic strategies.

Outer membrane vesicles have been implicated in antibiotic resistance in several bacterial species, primarily through mechanisms such as carrying enzymes that degrade antibiotics, binding and sequestering antimicrobial compounds, and facilitating horizontal gene transfer [36,37,38,39]. Antibiotic use itself can also increase OMV production, further enhancing bacterial survival and resistance. Although antibiotic resistance in *Borrelia burgdorferi* is less well studied, OMVs derived from this organism could similarly contribute to survival under antibiotic pressure by transporting proteins or DNA that may enhance bacterial survival [17]. Placing our findings alongside known OMV-mediated resistance pathways highlights how these vesicles may contribute to antibiotic responses in *B. burgdorferi*.

While direct infection with live *B. burgdorferi* spirochetes resulted in expression of both 16S rRNA and OspA, cells co-cultured with Bb-OMVs expressed only OspA, confirming vesicle uptake without bacterial contamination. These results support the idea that OMVs are capable of delivering bacterial proteins into mammalian cells and may influence host cell behavior even in the absence of live bacteria This aligns with previous suggestions that *B. burgdorferi* OMVs might serve as immune modulators, facilitate tissue invasion, or contribute to chronic disease persistence [40,41]. Currently, no standardized OMV isolation method exists that ensures complete recovery of all extracellular vesicle fractions while maintaining their native structure and functional properties [42,43,44].

In conclusion, our study demonstrates that the method of OMV isolation and the type of culture serum significantly impact OMV quality, size distribution, and cargo composition. Ultracentrifugation, particularly when paired with rabbit serum, offers a high-yield, well-characterized method for producing pure OMVs consistent with those described in the literature. Our results contribute to the growing understanding of *B. burgdorferi* OMVs and support their potential role in mediating host–pathogen interactions, with implications for bacterial persistence and possibly cancer progression. Furthermore, these findings could inform the development of improved diagnostics for Lyme disease by identifying OMV-associated biomarkers, such as OspA, that are readily detectable in patients’ samples. Additionally, understanding OMV composition and functionality may guide future therapeutic strategies aimed at modulating host–pathogen interactions to limit infection or inflammation.

## 4. Materials and Methods

### 4.1. Bacterial Cell Culture

The B31 strain of *Borrelia burgdorferi* (ATCC 35210, Manassas, VA, USA) was cultured in Barbour-Stonner-Kelly-H medium (BSK-H) supplemented with 6% rabbit serum (Pel-Freeze, Rogers, AR, USA). The bacterial cultures were grown in sterile 15 mL Falcon tubes (Thermo Scientific, Waltham, MA, USA) at 33 °C with 3% CO_2_ in a humidified environment. Low passage numbers (<P6) were used for all experiments.

### 4.2. Mammalian Cell Culture

The triple-negative breast cancer epithelial cells, MDA-MB-231 (ATCC HTB-26, Manassas, VA, USA), were grown using standard tissue culture conditions at 37 °C with 5% CO_2_ in a humidified environment using high glucose Dulbecco’s Modified Eagle Medium (DMEM, Gibco, ThermoFisher, Waltham, MA, USA) supplemented with 10% Fetal Bovine Serum (FBS, Sigma Aldrich, St. Louis, MO, USA), 1% Penicillin-Streptomycin (PS, Corning, Corning, NY, USA), 2 mM L-Glutamine (Sigma Aldrich, St. Louis, MO, USA).

### 4.3. Isolation and Purification of Bacterial Outer Membrane Vesicles

Two methods were used to isolate and purify outer membrane vesicles (OMVs) from *B. burgdorferi*, B31 strain (Figure 8). While ultracentrifugation has been shown to be highly effective and yield very pure vesicles [1,17,25,29], this isolation method requires specialized equipment and is less accessible for smaller laboratories. The ExoBacteria™ kit is more affordable, user-friendly, and accessible. Additionally, the effect of growth serum on Bb-OMV stimulation was compared between rabbit serum (Pel-Freeze, Rogers, AR, USA) used in culturing *B. burgdorferi* and exosome-depleted FBS (ED-FBS, Thermo Fisher Scientific, Waltham, MA, USA) (Table 2). The preparation of bacterial culture supernatant containing Bb-OMVs remained the same for both isolation methods. The bacterial cells were cultured in BSK-H medium supplemented with 6% rabbit serum till the mid-log phase growth (5 × 10^7^ Borrelia/mL density). Then 25 mL of cultures were pooled and spun at 3000 rpm for 15 min at room temperature to pellet the bacteria. The pellet was then resuspended in fresh BSK-H medium with 6% rabbit serum or 10% ED-FBS. The tubes were incubated at a higher temperature of 37 °C with a higher CO_2_ level of 5% in a humidified environment for 2 h to promote Bb-OMVs production. Then 20 mL of bacterial cultures per condition were pooled and spun at 3000 rpm for 20 min at 4 °C to pellet the bacterial cells. The supernatant was transferred to a new tube and spun down again to pellet any cellular debris. The supernatant was filtered through a 0.45 µm filter (Nalgene, Thermo Scientific, Waltham, MA, USA) followed by another filtration through a 0.22 µm filter (Nalgene, Thermo Scientific, Waltham, MA, USA).

The first method of isolation used the commercially available ExoBacteria™ OMV Isolation Kit (System Biosciences (SBI), Palo Alto, CA, USA) based on an ion-exchange chromatography system consisting of a binding resin and a gravity column to capture Bb-OMVs from a bacterial culture medium. The binding resin was placed into a column and equilibrated using the binding buffer in the kit. Then 25 mL of prepared bacterial supernatant was added to the resin and incubated at 4 °C on shaker conditions for 30 min, to allow the outer membrane vesicles to bind. The resin/supernatant mix was then allowed to flow through, and the resin was washed twice. The OMV elution buffer from the kit was incubated on top for two minutes. The eluate containing the Bb-OMVs was aliquoted in Eppendorf tubes and stored at −20 °C till further use.

The second method of isolation used was ultracentrifugation as previously described with modifications [17]. The 25 mL of prepared bacterial supernatant as described above was ultracentrifuged using polypropylene tubes (Beckman #355642, Brea, CA, USA) in a SW28 rotor of Beckman L-70M ultracentrifuge for 2 h at 100,000× *g*, 4 °C. The pellet was resuspended in PBS, aliquoted, and stored at −20 °C till further use.

### 4.4. Transmission Electron Microscopy

Transmission electron microscopy (TEM) was performed by Yale’s Electron Microscopy Facility. The negative staining of Bb-OMVs performed was similar to previously described methods [17]. The purified exosomes were diluted 1:20 in 1× PBS (pH 7.4). Then, a 7 µL droplet of the diluted sample was placed on a piece of parafilm and a glow-discharged 400 mesh carbon-coated Copper grid was placed on top of the droplet. This grid was incubated for 1 min. The remaining solution was gently blotted off the grid with a wedge of filter paper. This grid was then transferred to another 7 µL droplet of 2% uranyl acetate (aqueous) and incubated for another minute. The excess solution on the grid was gently blotted off again using a wedge of filter paper and the grid was allowed to air dry. The grids were imaged on a Tecnai G2 Spirit BioTWIN Transmission Electron Microscope (ThermoFisher Scientific, Hillsboro, OR, USA), which was operated at 80 kV. The TEM micrographs were acquired using a NanoSprint15 MKII camera (AMT Imaging, Woburn, MA, USA).

### 4.5. Western Blot

The bacterial cell pellets containing 10^8^ spirochetes were rinsed once with cold 1X PBS pH 7.4, lysed with NP-40 lysis buffer (J60766, Alfa Aesar, Ward Hill, MA, USA), and centrifuged at 12,000 rpm for 10 min at 4 °C. The total protein content of the Bb-OMV isolates and bacterial cell lysate was measured using the Bradford’s method (Bio-Rad, Hercules, CA, USA), according to the manufacturer’s recommendations and stored at −80 °C until further use. SDS-PAGE was performed by loading wells with 10 µg of *B. burgdorferi* cell lysate and Bb-OMV isolates. Precast 4–15% Mini-Protean TGX gels (Bio-Rad, Hercules, CA, USA) were run at 100 V for 1 h. Gels were transferred to Trans-Blot Turbo mini PVDF (Bio-Rad, Hercules, CA, USA) membranes using the Trans-Blot Turbo transfer system. After transfer, the membrane was washed once in Tris-buffered saline with Tween 20 (TBST), then incubated in 5% BSA (GoldBio, St. Louis, MO, USA) in TBST blocking solution for 1 h at RT. Membranes were washed three times for 10 min each in TBST buffer. Mouse monoclonal DnaK antibody (CB312) were donated by Dr. Jorge Benach of Stony Brook University and used at a dilution of 1:2 in TBST buffer supplemented with 5% BSA. Rabbit polyclonal OspA antibody (Rockland Immunochemicals, Rockland, MA, USA) was used at a dilution of 1:1000 in TBST buffer supplemented with 5% BSA. The membrane was cut between the wells and incubated in their respective primary antibody solutions overnight at 4 °C on a rocking plate shaker. Membranes were washed three times for 10 min each in TBST buffer. Membranes were incubated in respective Goat anti-Mouse HRP (Invitrogen, Waltham, MA, USA) and Goat anti-Rabbit HRP-conjugated (Abcam, Cambridge, UK) secondary antibodies at a dilution of 1:10,000. The membrane was incubated at RT for 1 h on a rocker. Membranes were washed three times for 10 min each in TBST buffer. Further, blots were incubated with 1:1 ECL chemiluminescent substrate and peroxidase solution (Bio-Rad, Hercules, CA, USA) for 3 min and visualized using Bio-Rad GelDoc Imaging System (Bio-Rad, Hercules, CA, USA) under chemiluminescence settings.

### 4.6. DNA and Protein Quantification

The presence of DNA in the purified OMVs from *B. burgdorferi* was quantified using a Qubit 2.0 fluorometer (Life Technologies, Carlsbad, CA, USA). The *B. burgdorferi* genomic DNA purified using GeneJet genomic DNA purification kit, Protocol D (Thermo Scientific, Waltham, MA, USA), was used as a positive control, and 1 mg/mL bovine serum albumin (BSA, Biorad, Hercules, CA, USA) was included as a negative control. Using the Qubit dsDNA broad range assay kit (2–1000 ng range, Invitrogen, Waltham, MA, USA) and a sample volume of 5 μL in 195 μL of Qubit working solution, the samples were analyzed as instructed by the manufacturer. All experiments were performed independently at least 3 times, with each independent experiment consisting of at least three replicates (N = 9).

The protein content was measured using Bradford’s method (Biorad, Hercules, CA, USA). Additionally, 1 mg/mL BSA was used as a positive control, and *B. burgdorferi* genomic DNA was used as a negative control. All experiments were performed independently at least 3 times, with each independent experiment consisting of at least three replicates (N = 9).

### 4.7. Infection of MDA-MB-231 with Spirochetes/OMVs

The TNBC MDA-MB-231 cells were seeded at a concentration of 1 × 10^6^ cells in 60 mm culture dishes and serum-deprived for 12–14 h in serum-free basal growth media supplemented with 1% Penicillin–Streptomycin, 2 mM L-Glutamine and 20 µg/mL Gentamicin sulfate. Infections with *B. burgdorferi* spirochetes at a multiplicity of infection (MOI) of 60 or exposure of 100 µg of Bb-OMVs, as determined by the total protein content, were conducted at different timepoints of 24 h, 48 h, and 72 h. All spirochetal infections and Bb-OMVs exposures of mammalian cells were performed in co-culture media composed of 2/3rd BSK-H medium with 6% rabbit serum (Pel-Freeze, Rogers, AR, USA) and 1/3rd basal growth medium (DMEM basal medium) without serum and antibiotics. Uninfected cells grown in co-culture media were used as a negative control. The MDA-MB-231 cells were extensively washed with 1X PBS after infection/exposure to remove any spirochetes/Bb-OMVs in the media to analyze the uptake/internalization by the mammalian cells.

### 4.8. Real-Time PCR

The cell pellets of the infected cells and their co-culture controls were collected using a standard cell culture trypsinization protocol and centrifugation at 2200 rpm for 10 min at room temperature (RT). The pellets were resuspended in phosphate-buffered saline (PBS, Sigma Aldrich, St. Louis, MO, USA), followed by centrifugation at 7000 rpm for 10 min at RT. Total RNA was isolated from these pellets using the Qiagen RNeasy Mini Kit (Qiagen, Hilden, Germany), following the manufacturer’s protocol. For reverse transcription reaction, 500 ng of the total RNA samples for each experimental condition was used and cDNA synthesis was performed using the Verso cDNA Synthesis Kit (Thermo Scientific, Waltham, MA, USA) in a T100 Thermal Cycler (Biorad, Hercules, CA, USA), following the manufacturer’s protocol with a mixture of random hexamers and anchored oligo-dT RNA primers in a 3:1 (*v*/*v*) ratio. The RNA and cDNA samples were quantified using a NanoDrop™ Lite (Thermo Scientific, Waltham, MA, USA) spectrophotometer.

Real-time PCR was performed using 2X iTaq™ Universal SYBR^®^ Green Supermix (Biorad, Hercules, CA, USA), 400 nM of primers, and about 1000 ng of sample in a CFX Opus 96 Real-Time PCR System (Biorad, Hercules, CA, USA). Primer sequences and PCR conditions were obtained from PrimerBank-MGH-PGA Harvard University webserver (https://pga.mgh.harvard.edu/primerbank/, accessed on 24 July 2024). To confirm the presence and absence of *B. burgdorferi,* specific 16S rRNA and OspA primers were used. Primer sequences used in this study are mentioned in Table 3. The reaction conditions were as follows: initial denaturation at 95 °C for 10 min, followed by 40 cycles of 95 °C for 10 s, 52 °C/60 °C annealing temperature for 20 s, 72 °C for 15 s, and a melt curve analysis at 55 °C for 5 s and 95 °C for 5 s. For quantitation, the obtained data was normalized against GAPDH (housekeeping gene), and the expression fold-change was calculated using the Livak method [45]. All RT-qPCR experiments were performed independently at least 3 times, with each independent experiment consisting of at least four replicates (N = 12).

### 4.9. Statistical Analysis

Statistical analysis was performed using Student’s *t*-test for paired samples (Microsoft Excel, Redmond, WA, USA) for all the techniques used. Statistical significance was determined based on the following thresholds: * *p*-values < 0.05, ** *p*-values < 0.01 and *** *p*-values < 0.001.

## 5. Conclusions

In conclusion, these results obtained in this study provide a comparative framework for optimizing *B. burgdorferi* OMV isolation and characterization. By identifying how isolation techniques and serum conditions affect OMV yield, purity, and functionality, this study advances the standardization of OMV research and lays the groundwork for future investigations into their roles in Lyme disease pathogenesis.

## Figures and Tables

**Figure 1 antibiotics-14-01079-f001:**
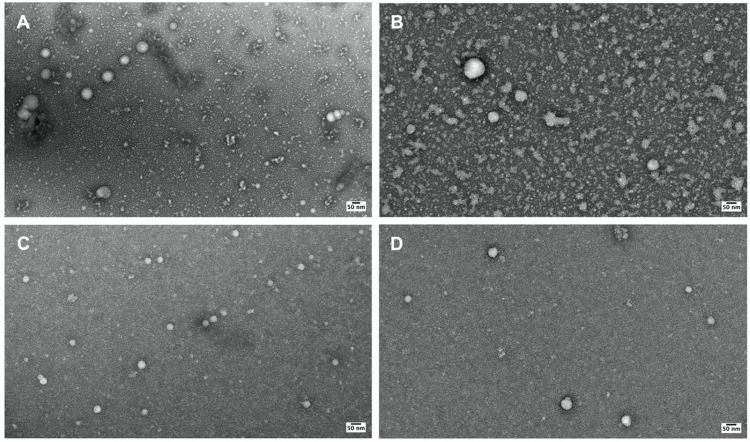
Transmission electron micrographs of purified OMVs from *B. burgdorferi*. (**A**,**B**) Micrograph of Bb-OMVs isolated using ultracentrifugation; (**C**,**D**) Micrograph of Bb-OMVs isolated using the ExoBacteria™ OMV Isolation Kit (System Biosciences (SBI)). Panel (**A**,**C**) shows results from isolation methods which used 6% rabbit serum and Panel (**B**,**D**) shows results from isolation methods which used exosome depleted fetal bovine serum. All images taken at 30,000× demonstrating spherical shapes with multiple single membrane Bb-OMVs; Scale bar: 50 nm.

**Figure 2 antibiotics-14-01079-f002:**
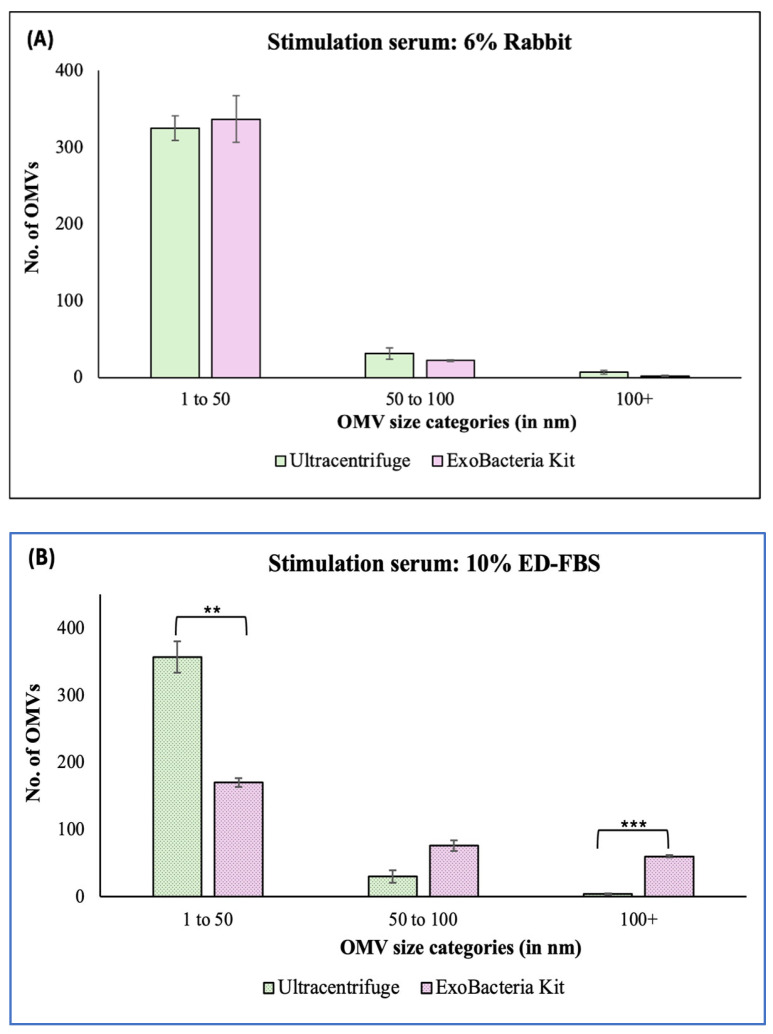
Size distribution of Bb-OMVs purified using different isolation methods and serum conditions. (**A**) Bb-OMVs isolated by ultracentrifugation or ExoBacteria™ kit following stimulation with 6% rabbit serum. (**B**) Bb-OMVs isolated by the same isolation methods with 10% ED-FBS stimulation. Diameters of 300 Bb-OMV vesicles of each experimental condition were measured using the TEM micrographs and grouped into 0–50 nm, 50–100 nm, and >100 nm. Statistical analysis was performed using Student’s *t*-test for paired samples (Microsoft Excel, Redmond, WA, USA). Statistical significance is indicated as ** *p* < 0.01, and *** *p* < 0.001.

**Figure 3 antibiotics-14-01079-f003:**
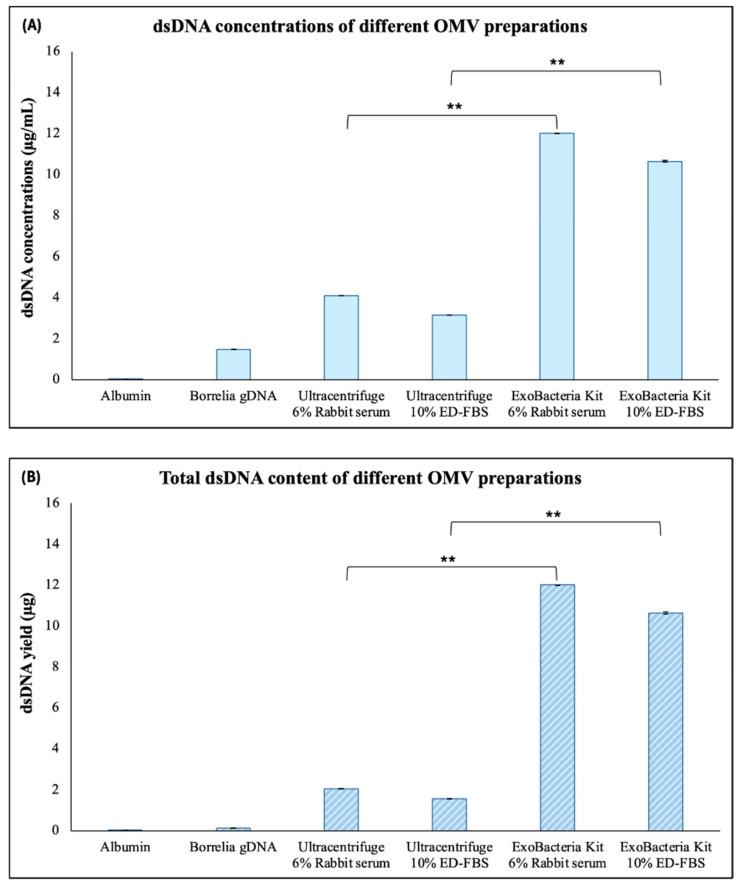
Quantification of dsDNA in different Bb-OMV samples isolated by ultracentrifugation and the ExoBacteria™ kit with different stimulation sera, using a Qubit 2.0 fluorometer with broad-range assay. *B. burgdorferi* B31 strain gDNA was used as a positive control and 1 mg/mL albumin was used as a negative control. (**A**) dsDNA concentrations (µg/mL) of different preparations of purified Bb-OMVs; (**B**) Total dsDNA content (µg) in different preparations of purified Bb-OMVs. The error bars indicate ± standard error of the mean (SEM) for each data point. The dsDNA content of Bb-OMVs of different isolation methods were compared. Statistical analysis was performed using Student’s *t*-test for paired samples (Microsoft Excel, Redmond, WA, USA). Statistical significance is indicated as ** *p* < 0.01.

**Figure 4 antibiotics-14-01079-f004:**
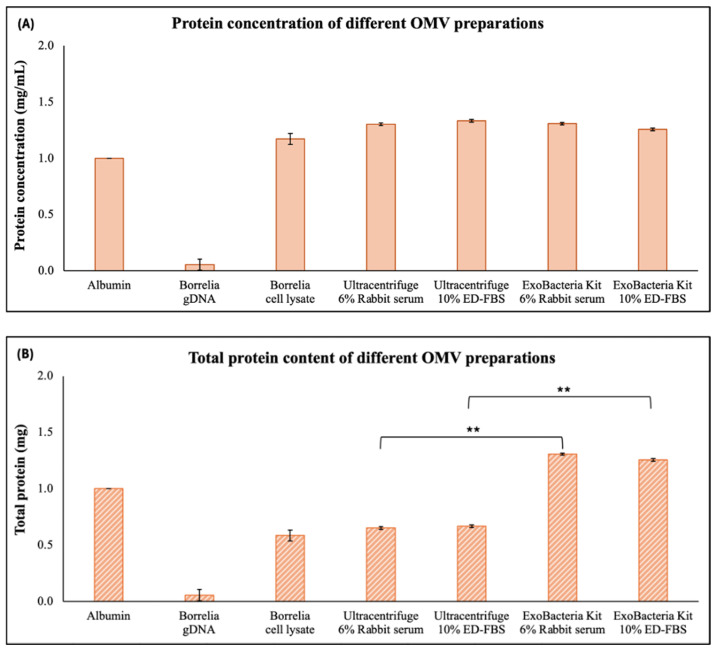
Quantification of protein in different Bb-OMV samples isolated by ultracentrifugation and the ExoBacteria™ kit with different stimulation sera, using Bradford’s assay. 1 mg/mL albumin was used as a positive control and *B. burgdorferi* B31 strain gDNA was used as a negative control. (**A**) protein concentrations (mg/mL) of different preparations of purified Bb-OMVs; (**B**) Total protein content (mg) in different preparations of purified Bb-OMVs. The error bars indicate ± standard error of the mean (SEM) for each data point. The protein content of Bb-OMVs of different isolation methods were compared. Statistical analysis was performed using Student’s *t*-test for paired samples (Microsoft Excel, Redmond, WA, USA). Statistical significance is indicated as ** *p* < 0.01.

**Figure 5 antibiotics-14-01079-f005:**
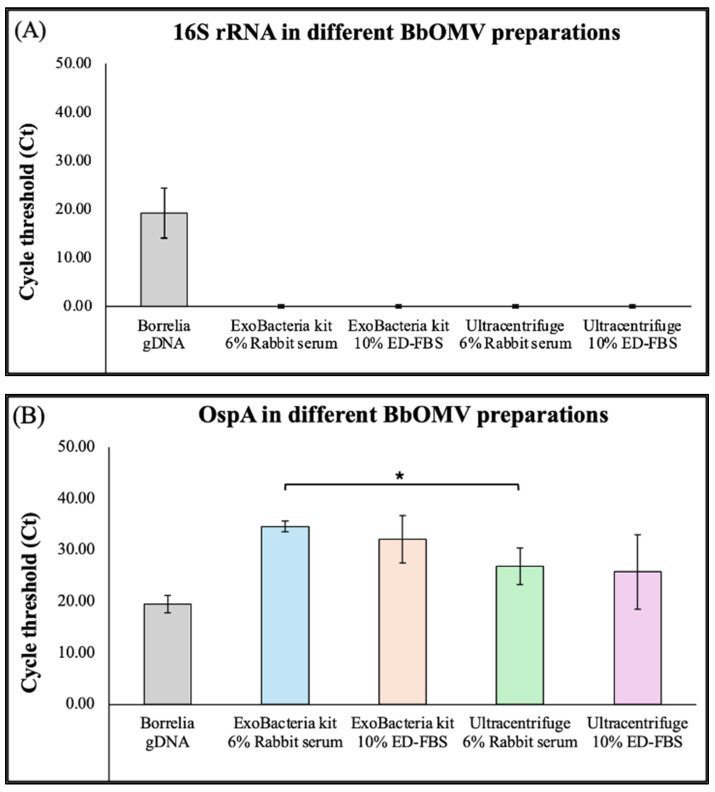
Gene expression profiles of OMVs isolated from *B. burgdorferi* using RT-PCR. Expression levels of (**A**) 16S rRNA bacterial signature and (**B**) OspA, outer surface protein in different Bb-OMV preparations. *B. burgdorferi* genomic DNA was used as a positive control. The results were compared between the OMV isolation methods using the same stimulation serum. The error bars indicate ± standard error of the mean (SEM) for each data point. Statistical analyses were performed using Student’s *t*-test for paired samples (Microsoft Excel, Redmond, WA, USA), on three independent RT-PCR experiments. Statistical significance is indicated as * *p* < 0.05.

**Figure 6 antibiotics-14-01079-f006:**
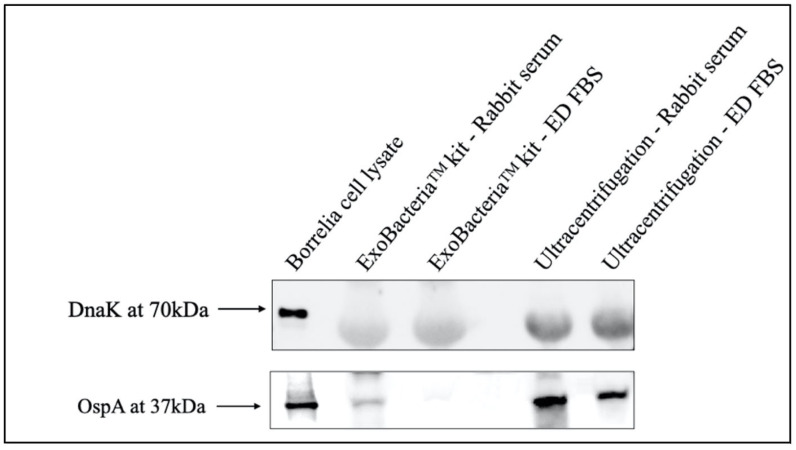
Evaluation of the protein expression profiles of the *B. burgdorferi* OMVs isolated with different methods using Western blot technique. The characteristic protein expression profiles of the Bb-OMV samples isolated via ultracentrifugation and ExoBacteria™ kit, were verified by the absence of the intracellular protein DnaK and the presence of membrane-specific OspA. *B. burgdorferi* strain B31 bacterial cell lysate was used as a positive control.

**Figure 7 antibiotics-14-01079-f007:**
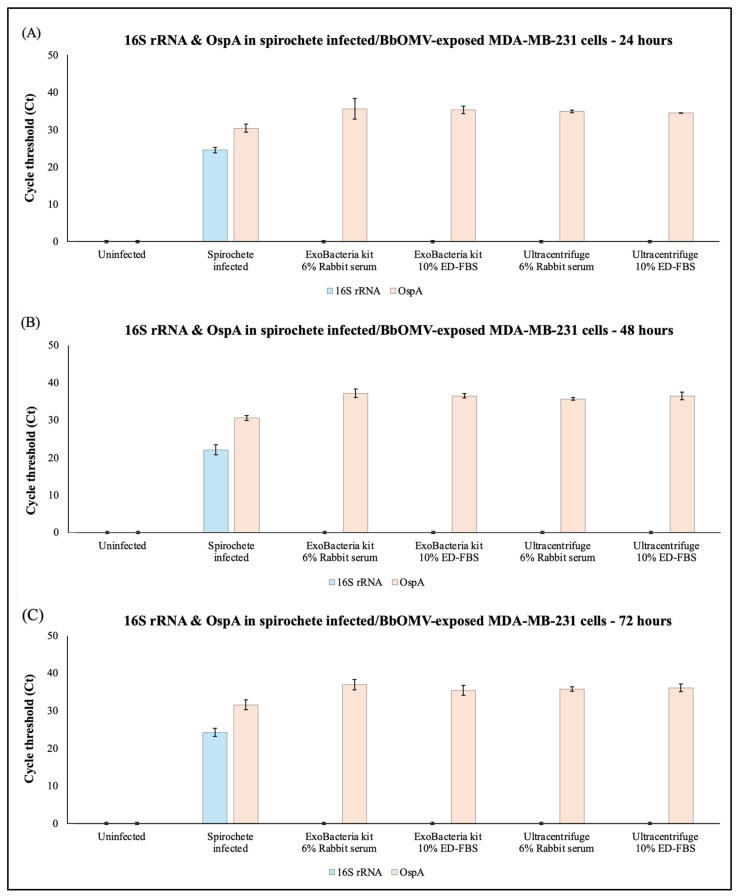
Evidence of spirochete and Bb-OMV uptake in MDA-MB-231 cells following infection with *B. burgdorferi* spirochetes or exposure to *B. burgdorferi* OMV samples for 24 h (Panel (**A**)), 48 h (Panel (**B**)) and 72 h (Panel (**C**)) using RT-qPCR method as described below. 16S rRNA was used as a marker for bacterial signature in infected or OMV-exposed MDA-MB-231 cells, while OspA indicates the presence of *B. burgdorferi*-specific outer surface membrane protein in infected or exposed MDA-MB-231 cells. The results were compared between the OMV isolation methods using the same stimulation serum. The error bars indicate ± standard error of the mean (SEM) for each data point. Statistical analyses were performed using Student’s *t*-test for paired samples (Microsoft Excel, Redmond, WA, USA), on three independent RT-qPCR experiments.

**Figure 8 antibiotics-14-01079-f008:**
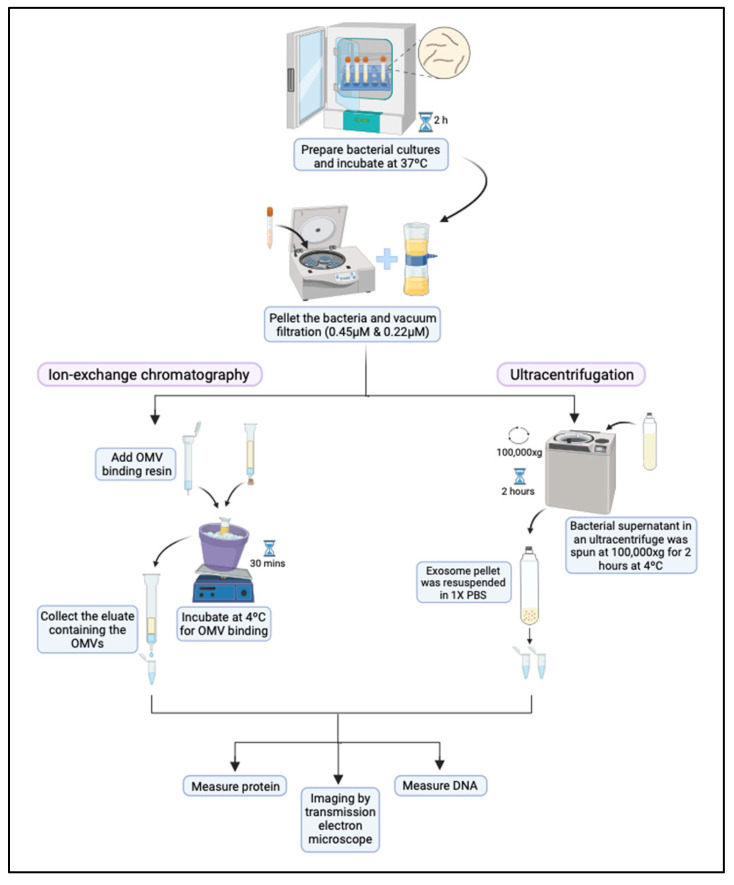
Schematic representation of different methods used to isolate and purify outer membrane vesicles (OMVs) from *B. burgdorferi*, B31 strain. Bacterial cultures were grown in BSK-H with 6% rabbit serum to mid-log phase, pelleted, resuspended in BSK-H with either 6% rabbit serum or 10% exosome-depleted FBS and incubated at 37 °C with 5% CO_2_ for 2 h to stimulate Bb-OMV release. Culture supernatants were clarified by centrifugation and sequential 0.45 µm/0.22 µm filtration. Bb-OMVs were isolated either using the ExoBacteria™ Kit (ion-exchange resin, SBI, Palo Alto, CA, USA) or by ultracentrifugation (100,000× *g*, 2 h, 4 °C), and stored at −20 °C.

**Table 1 antibiotics-14-01079-t001:** Fold change in OspA expression in MDA-MB-231 cells following 24 h, 48 h, and 72 h exposure to *B. burgdorferi* outer membrane vesicle (Bb-OMV) preparations. Comparison is made between OMVs isolated using the ExoBacteria kit and ultracentrifugation methods, under stimulation with two different sera. Expression levels were determined by RT-qPCR analysis. Statistical significance is denoted as * *p* < 0.05.

	6% Rabbit Serum				10% ED-FBS			
	Fold Change				Fold Change			
	Uninfected Control	ExoBacteria Kit	Ultracentrifuge	*p*-Value	Uninfected Control	ExoBacteria Kit	Ultracentrifuge	*p*-Value
24 h	0.0	1.0	1.9	0.54	0.0	1.0	1.7	0.18
48 h	0.0	1.0	4.0	0.01 (*)	0.0	1.0	1.2	0.67
72 h	0.0	1.0	1.8	0.22	0.0	1.0	0.5	0.22

**Table 2 antibiotics-14-01079-t002:** Summary of experimental conditions used for outer membrane vesicle (OMV) stimulation.

Method of Isolation	Accessibility	Stimulation Serum	Conditions for OMV Stimulation
Ultracentrifuge	Less accessible, specialized equipment	6% Rabbit serum	2 h at 37 °C with 5% CO_2_
		10% Exosome-depleted FBS	
ExoBacteria^TM^ kit	More accessible, commercially available, user-friendly	6% Rabbit serum	
		10% Exosome-depleted FBS	

**Table 3 antibiotics-14-01079-t003:** Sequences of primers used for RT-qPCR analysis.

Target Gene	Forward Primer (5′–3′)	Reverse Primer (5′–3′)	Annealing Temperature
16S rRNA	CCTGGCTTAGAACTAACG	CCTACAAAGCTTAATTCCTCAT	52 °C
OspA	GAACCAGACT-GAATACACAGGA	TTCAGCAGTTAGAGTTCCTTCA	60 °C
GAPDH	GGAGCGAGATCCCTCCAAAAT	GGCTGTTGTCATACTTCTCATGG	52 °C

Primer sequences are presented in the 5′–3′ direction.

## Data Availability

All data generated or analyzed during this study are available from the corresponding author upon reasonable request.

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
