# Peer review of "Evaluating Outer Membrane Vesicle Isolation Techniques for Borrelia burgdorferi and Their Impact on Vesicle Composition, Gene Expression Profile and Uptake"

_antibiotics, 2025, doi:10.3390/antibiotics14111079_

Round 1
Reviewer 1 Report
Comments and Suggestions for Authors
Summary:
Outer membrane vesicles are a relatively new area of research and their purification often requires access to an ultracentrifuge, which is often unavailable in smaller colleges. This is a very interesting concept overall - many smaller colleges don’t have access to an ultracentrifuge, but bacterial culture is simple enough for undergraduates to use. The authors investigate different isolation methods and culture additives to determine if OMVs can be isolated at high yield and purity without use of an ultracentrifuge. Further, the Borrelia-derived OMVs were analyzed for protein and DNA content, as well as their ability to enter mammalian cells.
General comments:
One major area of weakness is the MDA-MB-231 cell experiments (2.4). How do you know that the bacteria are inside of the mammalian cells through using PCR? Wouldn’t the cell lysis also lyse the bacteria, thereby releasing the RNA, even if the bacteria are outside of the cells? This experiment does not confirm that the bacteria (or OMVs) are inside of the MDA-MB-231 cells. Are the 24, 48, and 72 hours different treatments? Are all of the results the same? Overall, this section is very unclear regarding how the authors ran the experiments. Furthermore, the methods regarding culture and treatment of the MDA-MB-231 cells are missing in section 4.
As this is a special section on the Mechanism and Evolution of Antibiotic Resistance, it would be helpful to have discussion on how OMVs from Borrelia burgdorferi are related to antibiotic resistance in these species. Antibiotic resistance is not mentioned in the current article in either the introduction or discussion, which would both benefit from more context in this area or research.
Specific comments:
Lines 24 and 61 Purity was assessed with western blot? The terminology sounds incorrect - purity is usually assessed with SDS PAGE, and western blot is used to analyze if the correct protein is there, as western blot is a highly specific technique. Rephrasing this terminology such that it matches that used in the field of OMVs would make it clearer to the reader.
Line 26 to 28 - “To test their functionality” - as above, what functionality is being tested for? The ability of OMVs to enter cells is not their main function. Line 63 clarifies this, though, which this reviewer would suggest wording similarly here.
Line 87 - Is it missing punctuation, or is this a full sentence? It’s unclear what the authors are trying to say here.
Figure 2 - This figure took the reviewer a significant amount of time to understand what was being presented, as this is not normally how size distributions are shown. Most size data is shown as dynamic light scattering (DLS) data, which is plotted comparing the size and the number of OMVs. In this Figure, the reviewer is unsure what the categories add to the analysis, as they make it confusing for the reader to understand. This reviewer suggests reorganizing it as the number of OMVs on the y-axis and the diameter on the x axis. Alternatively, the authors could keep this, but remove the ‘total’ values since it adds little to the figure and confuses the reader, and instead put a chart for the averages
Figure 3 - Are the error bars really that small? Also, is there any significance between these groups? The authors state that t tests were done, with p values explained, but there are no p values shown on any figures in the paper. Also, what is the difference between A and B? It’s not described in the caption or the text.
Figure 4 - Again, what is the difference between A and B? Is it just mass compared to concentration? It is not described in the caption or the text.
Figure 5 and line 158 - “All OMV preparations tested positive for OspA,” - There is such little OspA for the Exobacteria kit with ED FBS that it’s nearly undetectable, and some might argue that it’s simply background or non-specific binding. The authors should rephrase their conclusion for this section.
Line 196 - Unclear what “biological interpretation” means here. More discussion needs to be provided.
Author Response
We greatly appreciate the time and effort you dedicated to reviewing our manuscript. Your thoughtful and constructive comments have been invaluable in strengthening our work. We have revised the manuscript accordingly and carefully addressed each of your suggestions. Our detailed responses to your feedback are provided below in red.
Outer membrane vesicles are a relatively new area of research and their purification often requires access to an ultracentrifuge, which is often unavailable in smaller colleges. This is a very interesting concept overall - many smaller colleges don’t have access to an ultracentrifuge, but bacterial culture is simple enough for undergraduates to use. The authors investigate different isolation methods and culture additives to determine if OMVs can be isolated at high yield and purity without use of an ultracentrifuge. Further, the Borrelia-derived OMVs were analyzed for protein and DNA content, as well as their ability to enter mammalian cells.
We thank the reviewer for the encouraging words regarding our work. As outer membrane vesicles (OMVs) remain a relatively new area of study, our research focuses on developing accessible isolation methods and characterizing Borrelia-derived OMVs to enable broader investigation even in settings without access to an ultracentrifuge.
General comments:
One major area of weakness is the MDA-MB-231 cell experiments (2.4). How do you know that the bacteria are inside of the mammalian cells through using PCR? Wouldn’t the cell lysis also lyse the bacteria, thereby releasing the RNA, even if the bacteria are outside of the cells? This experiment does not confirm that the bacteria (or OMVs) are inside of the MDA-MB-231 cells. Are the 24, 48, and 72 hours different treatments? Are all of the results the same? Overall, this section is very unclear regarding how the authors ran the experiments. Furthermore, the methods regarding culture and treatment of the MDA-MB-231 cells are missing in section 4.
Our previous studies have already demonstrated that Borrelia burgdorferi readily invades epithelial cells, supported by confocal microscopy1-3. In the OMV experiments, MDA-MB-231 cells were extensively washed after OMV exposure to remove any extracellular vesicles, and only then analyzed for OMV markers. Therefore, the PCR results reflect uptake/internalization by the mammalian cells rather than contamination from residual OMV in the medium.
- Wawrzeniak, K.; Gaur, G.; Sapi, E.; Senejani, A.G. Effect of Borrelia burgdorferi Outer Membrane Vesicles on Host Oxidative Stress Response. Antibiotics 2020, 9(5), 275.
- Gaur, G.; Sawant, J.Y.; Chavan, A.S.; Khatri, V.A.; Liu, Y.H.; Zhang, M.; Sapi, E. Effect of Invasion of Borrelia burgdorferiin Normal and Neoplastic Mammary Epithelial Cells. Antibiotics 2021, 10(11), 1295.
- Khatri, V.A.; Paul, S.; Patel, N.J.; Thippani, S.; Sawant, J.Y.; Durkee, K.L.; Murphy, C.L.; Ortiz, G.A.; Valentino, J.A.; Jathan, J.; Melillo, M.A.; Sapi, E. Global Transcriptomic Analysis of Breast Cancer and Normal Mammary Epithelial Cells Infected with Borrelia burgdorferi. J. Microbiol. Immunol. 2023, 13(3), 63–76.
Are the 24, 48, and 72 hours different treatments? Are all of the results the same? Overall, this section is very unclear regarding how the authors ran the experiments
The 24, 48, and 72-hour time points indeed represent different treatment conditions. We agree with the reviewer that presenting these results in a table format may have caused confusion; therefore, we have revised the presentation and now display the actual data as graphs (Figure 7). This format more clearly illustrates that the results are based on three independent experiments with a minimum sample size of four per condition (N=12), thereby strengthening the clarity and reproducibility of our findings.
Furthermore, the methods regarding culture and treatment of the MDA-MB-231 cells are missing in section 4.
We thank the reviewer for bringing this to our attention. The detailed description of the culture conditions and treatment of MDA-MB-231 cells has now been added to Section 4 of the revised manuscript to ensure clarity and reproducibility. Lines 369 to 375 and 481 to 494 in the revised manuscript.
As this is a special section on the Mechanism and Evolution of Antibiotic Resistance, it would be helpful to have discussion on how OMVs from Borrelia burgdorferi are related to antibiotic resistance in these species. Antibiotic resistance is not mentioned in the current article in either the introduction or discussion, which would both benefit from more context in this area or research.
We agree this is an important point and have incorporated a paragraph addressing it in the Discussion as follows:
“Outer membrane vesicles have been implicated in antibiotic resistance in several bacterial species, primarily through mechanisms such as carrying enzymes that degrade antibiotics, binding and sequestering antimicrobial compounds, and facilitating horizontal gene transfer [41-44]. Antibiotics use itself can also increase OMV production, further enhancing bacterial survival and resistance. Although antibiotic resistance in Borrelia burgdorferi is less well studied, OMVs derived from this organism could similarly contribute to survival under antibiotic pressure by transporting proteins or DNA that may enhance bacterial survival [17]. Placing our findings alongside known OMV-mediated resistance pathways highlights how these vesicles may contribute to antibiotic responses in B. burgdorferi.”
Lines 340 to 349 in the revised manuscript.
Specific comments:
Lines 24 and 61 Purity was assessed with western blot? The terminology sounds incorrect - purity is usually assessed with SDS PAGE, and western blot is used to analyze if the correct protein is there, as western blot is a highly specific technique. Rephrasing this terminology such that it matches that used in the field of OMVs would make it clearer to the reader.
We thank the reviewer for this clarification. We agree that the terminology could be more precise. In the revised manuscript, we have rephrased the description to indicate that SDS-PAGE was used to assess the purity of the OMV preparations, while Western blotting was employed specifically to confirm the presence of target OMV proteins. This aligns with standard terminology in the OMV field and should make the methods clearer to readers.
Lines 25 to 26 and line 85 in the revised manuscript.
Line 26 to 28 - “To test their functionality” - as above, what functionality is being tested for? The ability of OMVs to enter cells is not their main function. Line 63 clarifies this, though, which this reviewer would suggest wording similarly here.
We agree that the sentence in the abstract was confusing therefore we modified it:
"To assess their ability to interact with host cells, Bb-OMVs were co-cultured with MDA-MB-231 triple negative breast cancer cells. All isolated Bb-OMVs were taken up by the cells, as indicated by OspA expression, without detectable 16S rRNA, confirming vesicle internalization without bacterial contamination." Lines 28 to 31 in the revised abstract.
Line 87 - Is it missing punctuation, or is this a full sentence? It’s unclear what the authors are trying to say here.
We thank the reviewer for pointing this out. Indeed, the original text was missing punctuation and could be unclear. The sentence has been corrected in the revised manuscript. Line 111 in the revised manuscript.
Figure 2 - This figure took the reviewer a significant amount of time to understand what was being presented, as this is not normally how size distributions are shown. Most size data is shown as dynamic light scattering (DLS) data, which is plotted comparing the size and the number of OMVs. In this Figure, the reviewer is unsure what the categories add to the analysis, as they make it confusing for the reader to understand. This reviewer suggests reorganizing it as the number of OMVs on the y-axis and the diameter on the x axis. Alternatively, the authors could keep this, but remove the ‘total’ values since it adds little to the figure and confuses the reader, and instead put a chart for the averages
We thank the reviewer for the helpful feedback regarding Figure 2. While we used an established method to characterize Borrelia OMVs 4, we agree that the original presentation was not optimal and could be confusing. To improve clarity, we have completely revised Figure 2 to display the size distributions more intuitively, with OMV sizes on the x-axis and the number of OMVs on the y-axis, in line with standard practices in the field.
- Karvonen, K.; Tammisto, H.; Nykky, J.; Gilbert, L. Borrelia burgdorferi Outer Membrane Vesicles Contain Antigenic Proteins, but Do Not Induce Cell Death in Human Cells. 2022, 10(2), 212.
Figure 3 - Are the error bars really that small? Also, is there any significance between these groups? The authors state that t tests were done, with p values explained, but there are no p values shown on any figures in the paper. Also, what is the difference between A and B? It’s not described in the caption or the text.
We thank the reviewer for raising these points. The small error bars reflect the high reproducibility of our results, which remained consistent even with increased sample sizes. Panel A represents OMV concentration, while Panel B shows total yield; to improve clarity, we have added descriptive titles and adjusted the colors in Panel B. In addition, p-values indicating statistical significance have now been included on all relevant figures, in accordance with the methods described, to provide a clear representation of the differences between groups.
Figure 4 - Again, what is the difference between A and B? Is it just mass compared to concentration? It is not described. in the caption or the text
We thank the reviewer for pointing this out. In Figure 4, Panel A represents OMV protein concentration, while Panel B shows the total protein content in the different OMV samples. To clarify this distinction, we have updated the figure captions and added descriptive titles to each panel, consistent with the changes made for Figure 3, so that the differences are clear to the reader.
Figure 5 and line 158 - “All OMV preparations tested positive for OspA,” - There is such little OspA for the Exobacteria kit with ED FBS that it’s nearly undetectable, and some might argue that it’s simply background or non-specific binding. The authors should rephrase their conclusion for this section.
We agree with the reviewer. OspA was detected in all OMV preparations; however, signal levels were very low in the ExoBacteria kit with ED-FBS, approaching the limit of detection and potentially reflecting background or non-specific binding. Lines 214 to 216 in the revised manuscript.
Line 196 - Unclear what “biological interpretation” means here. More discussion needs to be provided.
We have revised the text to clarify the term “biological interpretation.” The sentence now reads: “Our findings emphasize the importance of carefully selecting both culture and isolation parameters, as these choices significantly affect the quality of OMV preparations and the conclusions that can be drawn about their composition, function, and interactions with host cells [17, 27].” Lines 264 to 267 in the revised manuscript. Also, additional discussion added to lines 304 to 319.
Reviewer 2 Report
Comments and Suggestions for Authors
OVERVIEW: In this technical and descriptive study, the authors compared two methods for the isolation of outer membrane vesicles (OMVs) from B. burgdorferi, the causative agent of Lyme disease. They focused their attention on two protocols: standard ultracentrifugation and the ion-exchange chromatography-based ExoBacteria™ kit, and examined how different serum supplements—rabbit serum and exosome-depleted FBS (ED-FBS)—influence OMV yield and properties. They concluded that both methods used give rise to intact and pure OMVs, as evidenced by the expression of outer surface protein A (OspA) and the presence of an intact membrane bilayer by TEM. In contrast, the DNA content is discriminating between the two approaches. Furthermore, the use of rabbit serum increased the yield of OMVs compared to those grown with ED-FBS. Moreover, the authors evaluated the functionality of OMVs by coculturing them with MDA-MB-231 cells and assessing their uptake by the cells.The authors aim to highlight the importance of choosing the appropriate method for OMV isolation.
The article can be accepted only after significant improvements in the data presented, the type of analysis performed, and the text in terms of arguments.
MAJOR COMMENTS:
Introduction:
The Introduction could highly benefit from the addition of details about B. burgdorferi's life cycle and the ability of B. burgdorferi spirochetes to invade cells by adding updated references about tumorigenic mammary epithelial cells.
Results:
Lines 70 to 72: Can the authors clarify whether the data produced are the result of a single round of purification under different conditions?
Lines 76 to 81 and Figure 2: The authors proceed with a series of observations that could be corroborated by adding statistical analysis concerning the experiment.
Figure 2: Can the authors report details about the statistical test used to evaluate their analyses in the figure legend?
Lines 84 to 85 and Figure 2: The authors aim to assess the influence of different serum supplements; in Figure 2, the authors plotted the data relative to rabbit serum separately from ED-FBS serum. Can we assume that the two protocols were conducted in parallel? Can the authors establish a single graph? Can the authors include the statistical analysis in the figure legend?
Line 109: BSA is indicated as Fraction V here. What about the other fractions? Can the authors explain?
Figure 3: The figure legend should distinguish panel A from panel B. Can the authors add some details? Can the authors include the statistical analysis in the figure legend?
Lines 111 to 113 and Figure 3: The authors concluded that significant amounts of dsDNA were detected in all purified OMV preparations, with higher concentrations observed in OMVs isolated using the ExoBacteria™ kit, without apparently indicating the statistical analysis.
Figure 4: The figure legend should distinguish panel A from panel B. Can the authors add some details? Can the authors include the statistical analysis in the figure legend?
Lines 114 to 117 and Figure 4: The authors concluded that the concentrations of B. burgdorferi whole cell lysate and OMVs isolated using ultracentrifugation were found to be comparable, while slightly higher concentrations were observed in the ExoBacteria™ kit isolation method, without apparently reporting the statistical support. Can the authors include the statistical analysis?
Lines 138 to 139: The authors assessed the purity of the OMV samples by evaluating 16S rRNA and OspA in different OMV samples using RT-qPCR without including RT-qPCR data. The article could highly benefit from the addition of RT-qPCR data/graphs including statistical analysis of the results from the three independent experiments.
Lines 149 to 151: The authors evaluated the purity of the different OMV samples by Western blot analysis of the presence of the B. burgdorferi-specific outer surface protein A (OspA) and the absence of the intracellular heat shock protein DnaK. Can the authors provide additional markers?
Lines 172 to 174: The authors assessed the functionality of OMVs isolated via different protocols by evaluating the expression of 16S rRNA and the surface protein OspA in MDA-MB-231 cells infected with B. burgdorferi or co-cultured with OMVs using RT-qPCR. The article could highly benefit from the addition of RT-qPCR data/graphs including statistical analysis of the results from three independent experiments.
Discussion:
Lines 188 to 191: The authors introduce the discussion by arguing for the importance of proper OMV isolation in cancer. Can the authors further argue for and justify this assumption?
Lines 237 to 238: The authors state in the final paragraph that they assessed several markers using Western blot and RT-qPCR, but they don't include several markers in the Results section: could the authors include these additional data or clarify the sentence?
MINOR COMMENTS:
Keywords:
Line 34: Not all of the keywords chosen appear to represent the article's content. Perhaps the authors could find more suitable ones to better highlight the approaches used.
Introduction:
Lines 49 to 54: The authors may wish to consider more up-to-date references (e.g. PMID: 38165613).
Line 63: The authors could specify that MDA-MB-231 are breast cancer cells and justify their use.
Figure 6A: Here the authors included a schematic workflow of different methods used to isolate and purify OMVs from B. burgdorferi. Can the authors describe the figure in the legend? This would benefit from it.
Comments on the Quality of English LanguageThe article (antibiotics-3852070) can be accepted only after significant improvements in the data presented, the type of analysis performed, and the text in terms of arguments.
Specifically, the authors claim, in the materials and methods section, to have performed each experiment in at least biological triplicate, but apparently fail to include the statistical data in the graphs for each experiment. Furthermore, the RT-qPCR data are not clearly displayed but only presented as tables, which appears inappropriate.
Author Response
OVERVIEW: In this technical and descriptive study, the authors compared two methods for the isolation of outer membrane vesicles (OMVs) from B. burgdorferi, the causative agent of Lyme disease. They focused their attention on two protocols: standard ultracentrifugation and the ion-exchange chromatography-based ExoBacteria™ kit, and examined how different serum supplements—rabbit serum and exosome-depleted FBS (ED-FBS)—influence OMV yield and properties. They concluded that both methods used give rise to intact and pure OMVs, as evidenced by the expression of outer surface protein A (OspA) and the presence of an intact membrane bilayer by TEM. In contrast, the DNA content is discriminating between the two approaches. Furthermore, the use of rabbit serum increased the yield of OMVs compared to those grown with ED-FBS. Moreover, the authors evaluated the functionality of OMVs by coculturing them with MDA-MB-231 cells and assessing their uptake by the cells.The authors aim to highlight the importance of choosing the appropriate method for OMV isolation.
The article can be accepted only after significant improvements in the data presented, the type of analysis performed, and the text in terms of arguments.
We sincerely thank you for the detailed and constructive review of our manuscript. We have carefully incorporated the suggested revisions and updated the manuscript to address all comments and recommendations. Our point-by-point responses to your feedback are provided below in red.
MAJOR COMMENTS:
Introduction:
The Introduction could highly benefit from the addition of details about B. burgdorferi's life cycle and the ability of B. burgdorferi spirochetes to invade cells by adding updated references about tumorigenic mammary epithelial cells.
We thank the reviewer for this suggestion. We have expanded the Introduction to include details on B. burgdorferi’s life cycle and its ability to invade mammalian cells, including tumorigenic mammary epithelial cells, with updated references to support these points. Lines 58 to 69 in the revised manuscript.
“B. burgdorferi is an obligate extracellular spirochete with a complex life cycle alternating between Ixodes tick vectors and mammalian hosts [19]. In ticks, spirochetes reside in the midgut and are transmitted to mammals during blood feeding [19]. Once in the mammalian host, B. burgdorferi disseminates through the bloodstream and extracellular matrices, colonizing multiple tissues. During this process, spirochetes can invade various cell types, including endothelial, epithelial, and tumorigenic mammary epithelial cells, facilitating persistence, evasion of host immune responses, and chronic infection [20-26]. Notably, B. burgdorferi produces OMVs throughout its life cycle, which are thought to play a role in host-pathogen interactions by delivering bacterial proteins and nucleic acids to host cells, modulating immune responses, and potentially aiding in colonization and survival [13-18]. Understanding these dynamics provides critical context for investigating the functional relevance of OMVs in infection and host tissue interactions.”
Results:
Lines 70 to 72: Can the authors clarify whether the data produced are the result of a single round of purification under different conditions?
We thank the reviewer for this question. All data presented were obtained from three independent OMV purifications for each of the four experimental conditions, ensuring reproducibility and consistency across replicates.
Lines 76 to 81 and Figure 2: The authors proceed with a series of observations that could be corroborated by adding statistical analysis concerning the experiment.
We thank the reviewer for this suggestion. To address this, we have revised Figure 2 for improved clarity and have included statistical analyses, with p-values now provided on all relevant comparisons to support the conclusions drawn from the data.
Figure 2: Can the authors report details about the statistical test used to evaluate their analyses in the figure legend?
We incorporated this statement into Figure 2 legend: Statistical analysis was performed using Student’s t-test for paired samples (Microsoft Excel, Redmond, WA, USA). Statistical significance is indicated as p < 0.05, *p < 0.01, and **p < 0.001. Lines 132 to 133 in the revised manuscript.
Lines 84 to 85 and Figure 2: The authors aim to assess the influence of different serum supplements; in Figure 2, the authors plotted the data relative to rabbit serum separately from ED-FBS serum. Can we assume that the two protocols were conducted in parallel? Can the authors establish a single graph? Can the authors include the statistical analysis in the figure legend?
We thank the reviewer for these comments. The experiments using rabbit serum and ED-FBS were indeed performed in parallel for all conditions. As mentioned above, we have completely revised Figure 2 to combine some data into a simplified graph, and statistical analyses with p values are now included in the figure legend to indicate significant differences between the two isolation methods.
Line 109: BSA is indicated as Fraction V here. What about the other fractions? Can the authors explain?
We thank the reviewer for this comment. Bovine Serum Albumin (BSA) “Fraction V” refers to the common preparation of albumin. It is the most widely available and standardized form of BSA for laboratory use. For this reason, BSA Fraction V is routinely used as and protein standard in biochemical and microbiological experiments.
Figure 3: The figure legend should distinguish panel A from panel B. Can the authors add some details? Can the authors include the statistical analysis in the figure legend?
We thank the reviewer for raising this important point. In Figure 3, Panel A depicts OMV DNA concentration, whereas Panel B represents the total DNA content of the different OMV samples. To clarify this distinction, we have updated the figure captions and added descriptive titles for each panel as well statistical analyses in the figure legend.
Lines 111 to 113 and Figure 3: The authors concluded that significant amounts of dsDNA were detected in all purified OMV preparations, with higher concentrations observed in OMVs isolated using the ExoBacteria™ kit, without apparently indicating the statistical analysis.
Statistical analyses have now been added to Figure 3, and p values are included in the figure legend to indicate the significance of differences in dsDNA concentrations between the different OMV preparations. Lines 158 to 166 in the revised manuscript.
Figure 4: The figure legend should distinguish panel A from panel B. Can the authors add some details? Can the authors include the statistical analysis in the figure legend?
In Figure 4, Panel A depicts OMV protein concentration, whereas Panel B represents the total protein content of the different OMV samples. To clarify this distinction, we have updated the figure captions and added descriptive titles for each panel, consistent with the modifications made to Figure 3, ensuring the differences are clear to the reader.
Lines 114 to 117 and Figure 4: The authors concluded that the concentrations of B. burgdorferi whole cell lysate and OMVs isolated using ultracentrifugation were found to be comparable, while slightly higher concentrations were observed in the ExoBacteria™ kit isolation method, without apparently reporting the statistical support. Can the authors include the statistical analysis?
Statistical analyses have now been added to Figure 4 as well, and p values are included in the figure legend to indicate the significance of differences in protein concentrations between OMV preparations. Lines 169 to 177 in the revised manuscript.
Lines 138 to 139: The authors assessed the purity of the OMV samples by evaluating 16S rRNA and OspA in different OMV samples using RT-qPCR without including RT-qPCR data. The article could highly benefit from the addition of RT-qPCR data/graphs including statistical analysis of the results from the three independent experiments.
Thank you for the suggestion. We have now included RT-PCR data with graphs and statistical analyses from the three independent experiments, which clearly demonstrate the consistency and reproducibility of our findings, providing strong support for OMV purity. Lines 193 to 201 in the revised manuscript.
Lines 149 to 151: The authors evaluated the purity of the different OMV samples by Western blot analysis of the presence of the B. burgdorferi-specific outer surface protein A (OspA) and the absence of the intracellular heat shock protein DnaK. Can the authors provide additional markers?
We thank the reviewer for this suggestion. In our study, we carefully selected 16S rRNA as a negative control and OspA as a positive marker, alongside DnaK as an additional control, because these have been consistently shown, in both our work and prior published studies, to reliably indicate OMV purity and content. While additional markers are available, the selected set represents the most established and widely accepted indicators for B. burgdorferi OMVs. We believe this set of markers is appropriate for the current study.
Lines 172 to 174: The authors assessed the functionality of OMVs isolated via different protocols by evaluating the expression of 16S rRNA and the surface protein OspA in MDA-MB-231 cells infected with B. burgdorferi or co-cultured with OMVs using RT-qPCR. The article could highly benefit from the addition of RT-qPCR data/graphs including statistical analysis of the results from three independent experiments.
We thank the reviewer for this suggestion. We have now included RT-PCR data presented as graphs, along with statistical analyses from three independent experiments, to clearly show the expression of 16S rRNA and OspA in MDA-MB-231 cells following infection with B. burgdorferi or co-culture with OMVs. Lines 247 to 256 in the revised manuscript.
Discussion:
Lines 188 to 191: The authors introduce the discussion by arguing for the importance of proper OMV isolation in cancer. Can the authors further argue for and justify this assumption?
We are grateful for the reviewer’s helpful suggestions.
We have expanded the discussion to emphasize that “Proper OMV isolation is critical for accurately assessing their biological effects, particularly in cancer research. Contaminants such as free bacterial proteins, nucleic acids, or cellular debris can confound functional assays, leading to misleading conclusions about OMV-mediated effects on cancer cell behavior. By carefully selecting and validating isolation methods, researchers can ensure that observed interactions, such as uptake, signaling, or modulation of host cell pathways, are specifically attributable to OMVs. This systematic approach is essential for understanding their potential role in tumor progression, host-pathogen interactions, and the development of OMV-based diagnostic or therapeutic strategies.” Lines 311 to 319 in the revised discussion.
Lines 237 to 238: The authors state in the final paragraph that they assessed several markers using Western blot and RT-qPCR, but they don't include several markers in the Results section: could the authors include these additional data or clarify the sentence?
We have clarified the sentence to specify the markers used: “OspA as a positive marker, 16S rRNA as a negative control, and DnaK as an additional control.” This provides more detailed information on how OMV identity and purity were assessed using Western blot and RT-PCR. Lines 320 to 321 in the revised manuscript.
MINOR COMMENTS:
Keywords:
Line 34: Not all of the keywords chosen appear to represent the article's content. Perhaps the authors could find more suitable ones to better highlight the approaches used.
Thank you for the suggestion. We revised our keywords for the followings:
Lyme disease, Borrelia burgdorferi, Bacterial virulence factors, Outer membrane vesicles (OMVs), Host-pathogen interactions, OMV isolation, Mammalian cell uptake
Introduction:
Lines 49 to 54: The authors may wish to consider more up-to-date references (e.g. PMID: 38165613).
Thank you for the suggestion. We updated significant numbers of our references. Marked with red in the revised manuscript.
Line 63: The authors could specify that MDA-MB-231 are breast cancer cells and justify their use.
MDA-MB-231 cells are a widely used human triple-negative breast cancer cell line. We selected this cell line because previous studies, including our own, have shown that it efficiently internalizes B. burgdorferi and OMVs, making it an ideal model to study host-pathogen interactions and OMV uptake in mammalian epithelial cells.
We added the followings in the revised manuscript:
“MDA-MB-231 triple-negative breast cancer cell line was selected because previous studies, including our own, have shown that it efficiently internalizes B. burgdorferi and bacterial OMVs, making it an ideal model to study host-pathogen interactions and OMV uptake in mammalian epithelial cells” Lines 228 to 232 in the revised manuscript.
References:
Gaur, G.; Sawant, J.Y.; Chavan, A.S.; Khatri, V.A.; Liu, Y.H.; Zhang, M.; Sapi, E. Effect of Invasion of Borrelia burgdorferi in Normal and Neoplastic Mammary Epithelial Cells. Antibiotics 2021, 10(11), 1295.
Khatri, V.A.; Paul, S.; Patel, N.J.; Thippani, S.; Sawant, J.Y.; Durkee, K.L.; Murphy, C.L.; Ortiz, G.A.; Valentino, J.A.; Jathan, J.; Melillo, M.A.; Sapi, E. Global Transcriptomic Analysis of Breast Cancer and Normal Mammary Epithelial Cells Infected with Borrelia burgdorferi. Eur. J. Microbiol. Immunol. 2023, 13(3), 63–76.
Laotee, S.; Arunmanee, W. Genetically Surface-Modified Escherichia coli Outer Membrane Vesicles Targeting MUC1 Antigen in Cancer Cells. Biotechnol. Rep. 2024, 44, e00854.
Figure 6A: Here the authors included a schematic workflow of different methods used to isolate and purify OMVs from B. burgdorferi. Can the authors describe the figure in the legend? This would benefit from it.
We thank the reviewer for this suggestion. We have updated the legend for Figure 6A, now it is Figure 8 to include a full description of the schematic workflow for the different OMV isolation and purification methods from B. burgdorferi, which we believe will improve clarity for the reader. Lines 416 to 422 in the revised manuscript.
Comments on the Quality of English Language
The article (antibiotics-3852070) can be accepted only after significant improvements in the data presented, the type of analysis performed, and the text in terms of arguments.
Specifically, the authors claim, in the materials and methods section, to have performed each experiment in at least biological triplicate, but apparently fail to include the statistical data in the graphs for each experiment. Furthermore, the RT-qPCR data are not clearly displayed but only presented as tables, which appears inappropriate.
We thank again the reviewer for the detailed feedback. To address these concerns, we have significantly revised the manuscript: all experiments are now clearly presented with appropriate statistical analyses and error bars reflecting at least three biological replicates. Additionally, RT-PCR data have been converted from tables into graphs, with statistical significance indicated, to improve clarity and readability. These changes strengthen the presentation, support the conclusions, and ensure that the data are transparent and reproducible.
Reviewer 3 Report
Comments and Suggestions for Authors
The manuscript entitled “Evaluating Outer Membrane Vesicle Isolation Techniques for Borrelia burgdorferi and Their Impact on Vesicle Composition, Purity and Uptake” The study could make a substantial contribution to our knowledge of Borrelia burgdorferi as one of the agents responsible for human Lyme disease. Nonetheless, a number of improvements would enhance the manuscript's readability, depth of analysis, and applicability.
Comments for authors:
Abstract:
Comment: Although the abstract is understandable, it might be strengthened to highlight the wider implications by mentioning the functional analysis of OMVs, namely their interaction with breast cancer cells.
Introduction:
Comment: Even though the introduction discusses OMVs in B. burgdorferi relatively sound, it can be explained further to compare the systematic method of isolations as it is the fundamental phase of the research.
Methods:
Comments: In the methodology the isolation methods are comprehensively elucidated, but it could be valuable to highlight the advantages and limitations of each method in terms of affordability, accessibility and ease of use. Further detail explanation on the western blotting and RtT-Qpcr will increase duplicability. Figure 6B is referred to as a figure in the text, but it appears to be a table in format and presentation. Please revise.
Results:
Comments: The results are clearly presented; there is need to provide summary figure of the important discoveries for TEM and would be supportive. There is also need to compare the purity of the OMVs isolated by the two methods quantitatively.
Discussions:
Comments: The discussion effectively incorporates the results but could be further expanded to discuss how the research might examine the likely effects of the composition of serum on the vesicles connections with host cells or their functional characteristics.
Conclusions:
Comments: The conclusion is brief but could be expanded to deliberate on the possible applications of these discoveries for diagnostic and therapeutic improvement for Lyme disease.
References:
Comments: strengthen work by including more recent references and minor typographical errors and varying formatting in references were also noted.
Author Response
The manuscript entitled “Evaluating Outer Membrane Vesicle Isolation Techniques for Borrelia burgdorferi and Their Impact on Vesicle Composition, Purity and Uptake” The study could make a substantial contribution to our knowledge of Borrelia burgdorferi as one of the agents responsible for human Lyme disease. Nonetheless, a number of improvements would enhance the manuscript's readability, depth of analysis, and applicability.
We would like to express our sincere gratitude for your careful review of our manuscript. Your insightful feedback has helped us improve both the clarity and quality of the work. We have revised the manuscript to reflect your recommendations and have provided detailed responses to each point below in red.
Comments for authors:
Abstract:
Comment: Although the abstract is understandable, it might be strengthened to highlight the wider implications by mentioning the functional analysis of OMVs, namely their interaction with breast cancer cells.
We thank the reviewer for this helpful suggestion. In response, we have updated the end of our abstract to better highlight the functional analysis of OMVs and their broader implications. The revised text now reads:
“These findings indicate that isolated OMVs are biologically active and capable of interacting with mammalian cells, highlighting their potential role in host-pathogen interactions and the broader relevance of OMVs in studying bacterial modulation of mammalian cell behavior. Overall, both isolation methods produced high-quality OMVs, with ultracentrifugation yielding slightly more pure vesicles, emphasizing the importance of selecting appropriate isolation methods and culture conditions for functional OMV studies.”
This revision clarifies the functional relevance of OMVs and strengthens the context for their interaction with mammalian cells. Lines 32 to 37 in the revised abstract.
Introduction:
Comment: Even though the introduction discusses OMVs in B. burgdorferi relatively sound, it can be explained further to compare the systematic method of isolations as it is the fundamental phase of the research.
We thank the reviewer for this suggestion and have revised the introduction to better emphasize the fundamental importance of systematically comparing OMV isolation methods in B. burgdorferi as follows:
“Despite growing interest in OMVs and their role in bacterial pathogenicity, relatively few studies have systematically compared OMV isolation methods in B. burgdorferi [15, 27, 28]. Most studies have relied on ultracentrifugation [13-17, 27, 29], which, while effective, requires specialized equipment that may not be accessible in smaller laboratories or universities. Systematic evaluation of different isolation techniques is critical because the choice of method can influence OMV yield, purity, and cargo composition, ultimately affecting downstream functional analyses [30]. To address this gap, in the present study, we evaluated two OMV isolation techniques: the standard ultracentrifugation method and a commercially available ExoBacteria™ OMV isolation kit. By comparing these approaches under controlled conditions, we aimed to identify reliable and accessible methods for producing biologically active OMVs suitable for functional studies in mammalian cell models. We also investigated the effect of OMV production by culturing B. burgdorferi with the standard rabbit serum and exosome-depleted fetal bovine serum (ED-FBS).”
Lines 70 to 82 in the revised manuscript.
Methods:
Comments: In the methodology the isolation methods are comprehensively elucidated, but it could be valuable to highlight the advantages and limitations of each method in terms of affordability, accessibility and ease of use. Further detail explanation on the western blotting and RtT-Qpcr will increase duplicability. Figure 6B is referred to as a figure in the text, but it appears to be a table in format and presentation. Please revise.
We are grateful for the reviewer’s helpful suggestions.
Advantages and limitations of isolation methods: We have added text to the Methods section highlighting the advantages and limitations of each OMV isolation approach as follows:
“While ultracentrifugation was shown that it is highly effective and yields very pure vesicles but requires specialized equipment and is less accessible for smaller laboratories. The ExoBacteria™ kit is more affordable, user-friendly, and accessible. Lines 379 to 382 in the revised manuscript.
Western blotting and RT-qPCR details: We have checked the methodological details for Western blotting and RT-qPCR to improve replicability, including descriptions of marker selection, antibody sources, reaction conditions, and quantification approaches. Lines 379 to 382 in the revised manuscript.
Figure 6B presentation: We have revised Figure 6B (now Figure 8) and Table 1 to ensure its format and presentation are consistent with the rest of the Tables. Lines 439 to 465 in the revised manuscript.
Results:
Comments: The results are clearly presented; there is need to provide summary figure of the important discoveries for TEM and would be supportive. There is also need to compare the purity of the OMVs isolated by the two methods quantitatively.
Thank you for your comments. We updated Figure 2 to clarify the differences for 4 methods which included quantitative data and statistical analyses.
Discussions:
Comments: The discussion effectively incorporates the results but could be further expanded to discuss how the research might examine the likely effects of the composition of serum on the vesicles connections with host cells or their functional characteristics.
We thank the reviewer for this insightful comment. We have expanded the Discussion to address how serum composition can influence OMV characteristics and their interactions with host cells by the followings:
"Different serum supplements, such as rabbit serum versus exosome-depleted fetal bovine serum (ED-FBS), can affect Bb-OMV yield, size, and cargo composition, including DNA and protein content. These differences may, in turn, modulate OMV uptake by mammalian cells and their functional effects on host-pathogen interactions. By highlighting these influences, we emphasize the importance of carefully selecting culture conditions to ensure consistent and biologically relevant OMV preparations for functional studies [17, 27, 30]." Lines 304 to 310 in the revised manuscript.
Conclusions:
Comments: The conclusion is brief but could be expanded to deliberate on the possible applications of these discoveries for diagnostic and therapeutic improvement for Lyme disease.
Thank you for the suggestion. We added these three sentences to the end of discussion.
Furthermore, these findings could inform the development of improved diagnostics for Lyme disease by identifying OMV-associated biomarkers, such as OspA, that are readily detectable in patients’ samples. Additionally, understanding OMV composition and functionality may guide future therapeutic strategies aimed at modulating host-pathogen interactions to limit infection or inflammation.
Lines 356 to 361 in the revised manuscript.
References:
Comments: strengthen work by including more recent references and minor typographical errors and varying formatting in references were also noted.
We thank the reviewer for this helpful feedback. We have updated the manuscript to include more recent references where appropriate, corrected typographical errors, and standardized the formatting of all references to ensure consistency throughout the text.
Round 2
Reviewer 1 Report
Comments and Suggestions for Authors
Edits greatly improved the article. This reviewer has some additional comments for the authors below.
- The reviewer’s point regarding purity and western blot (Lines 24 and 85) was that the authors did not analyze purity as it is conventionally described, they analyzed if OspA was present and intracellular markers for Bb were absent. This is not how ‘purity’ is normally defined for OMVs in the conventional sense. It is unclear what the authors are defining purity in these lines. Section 2.3 is clearer on the author’s definition of purity, but the references to SDS PAGE should be removed (as the authors did not actually do an SDS PAGE analysis). The authors’ definition of purity in the abstract and introduction needs to be described much more clearly, and the references to SDS PAGE and purity should be removed here too.
- Figure 2 is significantly clearer in terms of the analysis, but the authors state in lines 101-102 “The sizes of purified B. burgdorferi OMVs from each isolation condition were analyzed from the transmission electron micrographs, by measuring the diameters of ~300 vesicles using ImageJ”. The number of vesicles in Figure 2B for the Exobacteria kit does not meet this threshold (nor did it in the first version of this figure). 300 vesicles should be analyzed for this group as well, in order to make a direct comparison between groups.
- Line 244-245 is an incomplete sentence.
- Lines 320-322 is an incomplete sentence.
- The addition of lines 306-319 is helpful to understanding the purpose of the cell studies, but should be moved closer to line 329, where the breast cancer studies are discussed, and transitioned appropriately.
- Lines 340-349 better address the special section on antibiotic resistance, but this paragraph should be mentioned earlier in the discussion, and transition into the work that the authors did in this study.
- The discussion should have some mention of the differences in ease and accessibility between ultracentrifugation and the exo-bacteria kit, which is a main component of the introduction, but is not currently included in the discussion. Since the multiple comparisons done in this study are very clearly shown in Table 1, the reviewer also suggests moving Table 1 to the discussion for the authors to reference.
Author Response
Edits greatly improved the article. This reviewer has some additional comments for the authors below.
We would like to sincerely thank the reviewer for the appreciation of our work and for the additional comments that will help improve our work further. We have incorporated the necessary changes to our manuscript as you suggested (marked with blue color). Below, we have included our detailed responses.
- The reviewer’s point regarding purity and western blot (Lines 24 and 85) was that the authors did not analyze purity as it is conventionally described, they analyzed if OspA was present and intracellular markers for Bb were absent. This is not how ‘purity’ is normally defined for OMVs in the conventional sense. It is unclear what the authors are defining purity in these lines. Section 2.3 is clearer on the author’s definition of purity, but the references to SDS PAGE should be removed (as the authors did not actually do an SDS PAGE analysis). The authors’ definition of purity in the abstract and introduction needs to be described much more clearly, and the references to SDS PAGE and purity should be removed here too.
We thank the reviewer for their insightful feedback. We deleted the word SDS-PAGE and replaced the word of “purity” with the word of “gene expression profiles or protein expression profiles.” Lines 4, 25, 86, 180, 197, 204, 221 and 321 in the revised manuscript.
- Figure 2 is significantly clearer in terms of the analysis, but the authors state in lines 101-102 “The sizes of purified B. burgdorferi OMVs from each isolation condition were analyzed from the transmission electron micrographs, by measuring the diameters of ~300 vesicles using ImageJ”. The number of vesicles in Figure 2B for the Exobacteria kit does not meet this threshold (nor did it in the first version of this figure). 300 vesicles should be analyzed for this group as well, in order to make a direct comparison between groups.
We thank the reviewer for highlighting this point. In response, we have updated Figure 2 in the revised manuscript to present an analysis of 300 vesicles for each experimental condition, as suggested. Line 132 in the revised manuscript.
- Line 244-245 is an incomplete sentence.
We agree with the reviewer that the sentence is incomplete. We have rewritten the sentence to incorporate the necessary changes.
“There were no statistically significant differences (p > 0.05) observed in the expression of OspA in the breast cancer cells exposed to different OMV sample preparations.” Lines 248-248 in the revised manuscript.
- Lines 320-322 is an incomplete sentence.
We thank the reviewer for pointing this out. The sentence has been corrected in the revised manuscript. Lines 321-323 in the revised manuscript.
- The addition of lines 306-319 is helpful to understanding the purpose of the cell studies, but should be moved closer to line 329, where the breast cancer studies are discussed, and transitioned appropriately.
We thank the reviewer for the feedback. We agree that the manuscript reads much better when lines 306-319 are addressed after line 329. We have moved the section starting from line 304-319 to after line 329. Lines 331-346 in the revised manuscript.
- Lines 340-349 better address the special section on antibiotic resistance, but this paragraph should be mentioned earlier in the discussion, and transition into the work that the authors did in this study.
We thank the reviewer for their insightful feedback and agree with their opinion. We have moved lines 340-349 to be earlier in the discussion section. Line 347-356 in the revised manuscript.
- The discussion should have some mention of the differences in ease and accessibility between ultracentrifugation and the exo-bacteria kit, which is a main component of the introduction, but is not currently included in the discussion. Since the multiple comparisons done in this study are very clearly shown in Table 1, the reviewer also suggests moving Table 1 to the discussion for the authors to reference.
We thank the reviewer for pointing this out. We agree that the discussion does not highlight the key point of the need for alternate OMV isolation methods, which forms the basis of our manuscript. We have addressed this issue and have also moved Table 1 to the discussion (Table 2 in the revised manuscript). Lines 271-276, 279-283 in the revised manuscript.
Reviewer 2 Report
Comments and Suggestions for Authors
In this technical and descriptive study, the authors compared two methods for the isolation of outer membrane vesicles (OMVs) from B. burgdorferi, the causative agent of Lyme disease. They focused their attention on two protocols: standard ultracentrifugation and the ion-exchange chromatography-based ExoBacteria™ kit, and examined how different serum supplements—rabbit serum and exosome-depleted FBS (ED-FBS)—influence OMV yield and properties. They concluded that both methods used give rise to intact and pure OMVs, as evidenced by the expression of outer surface protein A (OspA) and the presence of an intact membrane bilayer by TEM. In contrast, the DNA content is discriminating between the two approaches. Furthermore, the use of rabbit serum increased the yield of OMVs compared to those grown with ED-FBS. Moreover, the authors evaluated the functionality of OMVs by coculturing them with MDA-MB-231 cells and assessing their uptake by the cells. The authors aim to highlight the importance of choosing the appropriate method for OMV isolation.
The authors have comprehensively addressed and corrected the scientific manuscript text, performing thorough and detailed work. Nevertheless, one note/suggestion remains from the first round of review, which pertains to the RT-qPCR data presented. The article can be accepted only after improvements of data elaboration.
MAJOR COMMENTS:
RT-qPCR results are interesting! To make the biological significance even more apparent to readers, authors might consider presenting the data as fold changes (2^-ΔΔCt) alongside the Ct values. This approach, recommended in recent guidelines, would help readers immediately understand the magnitude of expression differences between your conditions. The fold change presentation is particularly useful for statistical analysis and cross-study comparisons.
Author Response
We would like to sincerely thank the reviewer for the additional comments that will help improve our work further. We have incorporated the necessary changes to our manuscript as you suggested (marked with blue color). Below, we have included our detailed responses.
In this technical and descriptive study, the authors compared two methods for the isolation of outer membrane vesicles (OMVs) from B. burgdorferi, the causative agent of Lyme disease. They focused their attention on two protocols: standard ultracentrifugation and the ion-exchange chromatography-based ExoBacteria™ kit, and examined how different serum supplements—rabbit serum and exosome-depleted FBS (ED-FBS)—influence OMV yield and properties. They concluded that both methods used give rise to intact and pure OMVs, as evidenced by the expression of outer surface protein A (OspA) and the presence of an intact membrane bilayer by TEM. In contrast, the DNA content is discriminating between the two approaches. Furthermore, the use of rabbit serum increased the yield of OMVs compared to those grown with ED-FBS. Moreover, the authors evaluated the functionality of OMVs by coculturing them with MDA-MB-231 cells and assessing their uptake by the cells. The authors aim to highlight the importance of choosing the appropriate method for OMV isolation.
The authors have comprehensively addressed and corrected the scientific manuscript text, performing thorough and detailed work. Nevertheless, one note/suggestion remains from the first round of review, which pertains to the RT-qPCR data presented. The article can be accepted only after improvements of data elaboration.
We would sincerely like to thank the reviewer for your praise and constructive feedback on our manuscript. We have revised the manuscript to include the necessary details that you have asked for and provided our responses below.
MAJOR COMMENTS:
RT-qPCR results are interesting! To make the biological significance even more apparent to readers, authors might consider presenting the data as fold changes (2^-ΔΔCt) alongside the Ct values. This approach, recommended in recent guidelines, would help readers immediately understand the magnitude of expression differences between your conditions. The fold change presentation is particularly useful for statistical analysis and cross-study comparisons.
We would like to thank the reviewer for the constructive feedback and the encouraging words towards our work. We agree with your suggestion and have included the data showing the log fold changes from our RT-qPCR experiments, along with the appropriate statistical analysis (Table 1 in the revised manuscript, lines 262-267). We also updated the method section with the necessary details about the quantitative analyses (lines 532-533 and Table 3)
We believe that the revisions made in response to the reviewers' comments have significantly improved our manuscript. The changes address all the concerns raised while maintaining the scientific rigor and novelty of our work.
We thank the reviewers once again for their valuable feedback and look forward to your decision.
Round 3
Reviewer 2 Report
Comments and Suggestions for Authors
In this technical and descriptive study, the authors compared two methods for the isolation of outer membrane vesicles (OMVs) from B. burgdorferi, the causative agent of Lyme disease. They focused their attention on two protocols: standard ultracentrifugation and the ion-exchange chromatography-based ExoBacteria™ kit, and examined how different serum supplements—rabbit serum and exosome-depleted FBS (ED-FBS)—influence OMV yield and properties. They concluded that both methods used give rise to intact and pure OMVs, as evidenced by the expression of outer surface protein A (OspA) and the presence of an intact membrane bilayer by TEM. In contrast, the DNA content is discriminating between the two approaches. Furthermore, the use of rabbit serum increased the yield of OMVs compared to those grown with ED-FBS. Moreover, the authors evaluated the functionality of OMVs by coculturing them with MDA-MB-231 cells and assessing their uptake by the cells. The authors aim to highlight the importance of choosing the appropriate method for OMV isolation.
The authors reviewed the presentation of the RT-qPCR data. Responsiveness to suggestions is appreciated.
MINOR COMMENTS:
There is a discrepancy between the labeling of the graphs and the diagrams, and between the labeling of the graphs and the new RT-qPCR table. Calculating 2^-ΔΔCt produces linear, not logarithmic, fold change values. Therefore, the y-axis or table should be labeled "Fold Change" or "Relative Expression (2^-ΔΔCt)" rather than "Logarithmic Fold Change." If authors prefer to present data on a logarithmic scale for better visualization, they will likely standardize the graphs and table.
Author Response
We are deeply grateful to the reviewer for the thoughtful and constructive feedback provided throughout the review process. The reviewer’s excellent suggestions and insightful comments have substantially strengthened the quality, clarity, and overall rigor of our manuscript.
As suggested, we have revised the wording in Table 1 to “fold change” (Lines 262–267 in the revised manuscript).